


# Concerted measurements of lipids in seawater and on submicron aerosol particles at the Cape Verde Islands: biogenic sources, selective transfer and high enrichments

Nadja Triesch[1], Manuela van Pinxteren[1], Sanja Frka[2], Christian Stolle[3,4], Tobias Spranger[1], Erik Hans Hoffmann[1], Xianda Gong[1], Heike Wex[1], Detlef Schulz-Bull[3], Blaženka Gašparović[2] and Hartmut Herrmann[1*]

[1]Leibniz-Institute for Tropospheric Research (TROPOS), Leipzig, 04318, Germany
[2]Division for Marine and Environmental Research, Rudjer Boskovic Institute, Zagreb, 100000, Croatia
[3]Leibniz-Institute for Baltic Sea Research Warnemuende (IOW), Rostock, 18119, Germany
[4]Institute for Chemistry and Biology of the Marine Environment (ICBM), Carl-von-Ossietzky University Oldenburg, Wilhelmshaven, 26382, Germany

*Correspondence to*: Hartmut Herrmann (herrmann@tropos.de)

## Abstract

Measurements of lipids as representative species for different lipid classes in the marine environment have been performed to characterize their oceanic sources and their transfer from the ocean into the atmosphere to marine aerosol particles. To this end, a set of lipid classes (hydrocarbons (HC), fatty acid methyl esters (ME), free fatty acids (FFA), alcohols (ALC), 1,3-diacylglycerols (1,3 DG), 1,2-diacylglycerols (1,2 DG), monoacylglycerols (MG), wax esters (WE), triacylglycerols (TG), phospholipids (PP) including phosphatidylglycerols (PG), phosphatidylethanolamine (PE), phosphatidylcholines (PC), glycolipids (GL) including sulfoquinovosyldiacylglycerols (SQDG), monogalactosyl-diacylglycerols (MGDG), digalactosyldiacylglycerols (DGDG) and sterols (ST)) is investigated in both the dissolved and particulate fraction in seawater, differentiated between underlying water (ULW) and the sea surface microlayer (SML), and in ambient submicron aerosol particle samples ($PM_1$) at the Cape Verde Atmospheric Observatory (CVAO) applying concerted measurements. The different lipids are found in all marine compartments but in different compositions. At this point, a certain variability is observed for the concentration of dissolved ($\sum DL_{ULW}$: 39.8-128.5 µg $L^{-1}$, $\sum DL_{SML}$: 55.7-121.5 µg $L^{-1}$) and particulate ($\sum PL_{ULW}$: 36.4-93.5 µg $L^{-1}$, $\sum PL_{SML}$: 61.0-118.1 µg $L^{-1}$) lipids in seawater of the tropical North Atlantic Ocean along the campaign. Only slight SML enrichments are observed for the lipids with an enrichment factor $EF_{SML}$ of 1.1-1.4 (DL) and 1.0-1.7 (PL). On $PM_1$ aerosol particles, a total lipid concentration between 75.2-219.5 ng $m^{-3}$ (averaged: 119.9 ng $m^{-3}$) is measured with high atmospheric concentration of TG (averaged: 21.9 ng $m^{-3}$) as a potential indicator for freshly emitted sea spray. Besides phytoplankton sources, bacteria influence the lipid concentrations in seawater and on the aerosol particles, so that the phytoplankton tracer (chlorophyll-*a*) cannot sufficiently explain the lipid abundance. The concentration and enrichment of lipids in the SML is not related to physicochemical properties describing the surface activity. For aerosol, however, the high enrichment of lipids (as a sum) on aerosols corresponds well with the consideration of their high surface activity, thus the $EF_{aer}$





(enrichment factor on submicron aerosol particles compared to SML) ranges between $9 \cdot 10^4$-$7 \cdot 10^5$. Regarding the single lipid groups on the aerosol particles, a weak relation between $EF_{aer}$ and lipophilicity (expressed by the $K_{OW}$ value) was identified, which was absent for the SML. However, overall simple physico-chemical descriptors are not sufficient to fully explain the transfer of lipids. As our findings show that additional processes such as formation and degradation influence the ocean-atmosphere transfer of both OM in general and of lipids in particular, they have to be considered in OM transfer models. Moreover, our data suggest that the extend of enrichment of lipid classes constituents on the aerosol particles might be related to the distribution of the lipid within the bubble-air-water-interface. Lipids, which are preferably arranged within the bubble interface, namely TG and ALC, are transferred to the aerosol particles to the highest extend. Finally, the connection between ice nucleation particles (INP) in seawater, which are active already at higher temperatures (-10 °C to -15 °C), and the lipid classes PE and FFA suggests that lipids formed in the ocean have the potential to contribute to (biogenic) INP activity when transferred to the atmosphere.

**Keywords**

Lipids, organic matter, submicron aerosol particles ($PM_1$), seawater, biogenic sources, ice nucleating particles (INP), transfer, enrichment factor, concerted measurements, Cape Verde Atmospheric Observatory (CVAO)

## 1.   Introduction

Lipids are a major biochemical class of organic matter (OM) in seawater along with carbohydrates and proteins. Their ocean concentrations are much lower, yet their surface affinity and enrichment are higher than for the other groups (Burrows et al., 2014). As lipids are rich in carbon and serve as energy storage compounds, they are important components of the cellular metabolisms of species, at least in the ocean (Wakeham et al., 1997). They are universally distributed in the marine environment and are involved in numerous essential biological processes of both the dissolved and particulate OM pool (Arts et al., 2001;Frka et al., 2011). As regards the particulate lipid pool, important sources are phytoplankton cells with a contribution of up to 79 % in biologically productive surface water layers, while the contribution of lipids in phytoplankton cells ranges between $\leq$ 1 % and 46 % of dry weight (Frka et al. (2011), and references therein). Marine dissolved lipids are produced either by dissolution from the particulate fraction, or through primary production and could be released during life cycle or after cell death (Yoshimura et al., 2009;Novak et al., 2018). As a whole, the analysis of lipids and their classes is a useful method to study the dynamics of the global carbon cycle in the ocean (Yoshimura et al., 2009), since lipid classes can be used as specific markers for the identification of OM sources and biogeochemical cycles in the marine environment (Parrish et al., 1988;Frka et al., 2011). Phosphorus-containing lipids or phospholipids (PP), i.e. phosphatidylglycerols (PG), phosphatidylethanolamine (PE) and phosphatidylcholines (PC), belong to the organic substances associated with living organisms (Derieux et al., 1998) since they are a major component of cell membranes providing the structure and protection of cells (Khozin-Goldberg, 2016). The most common glycolipids (GL) in plankton are mono- and di-galactosyldiacylglycerols





(MGDG and DGDG) as well as sulfoquinovosyldiacylglycerol (SQDG) (Guschina and Harwood, 2009;Gašparović et al., 2013). Triacylglycerols (TG) indicate metabolic reserves (Frka et al., 2011) and wax esters (WE) are a major group of neutral lipids of some zooplankton species (Kattner, 1989). Fatty alcohols (ALC) mainly originate from zooplankton wax esters (Frka et al., 2011). Diacylglycerides (DG), monoacylglycerides (MG) and free fatty acids (FFA) are glyceride degradation products

which characterize the degradation level of lipids by means of the lipolysis index (LI) (Parrish et al., 1988;Goutx et al., 2003). Using lipids as biomarkers, estimates of marine OM sinks and sources can be made and previous OM parameterizations, based only on chl-*a* can be extended. Indeed, chl-*a* is often used as an indicator of marine biological productivity, representing the abundance of the major group of organisms, i.e. the photoautotrophs (Gantt et al., 2011;Rinaldi et al., 2013). However, chl-*a* is also found to be a poor descriptor of autotrophs, especially in oligotrophic regions (Quinn et al., 2014). To fully describe

the biological control of the OM cycle, both autotrophic and heterotrophic organisms must be considered. The information on the abundance of the main OM classes, namely lipids, proteins and carbohydrates, contributing to the marine OM pool and reflecting the OM sources, can therefore be used for advanced modelling approaches to predict OM on marine aerosol particles depending on the chemical composition of marine OM, as e.g. described by Burrows et al. (2014). Further spatio-temporal investigations of the organic classes in different oceanic areas are important for this purpose. The study of lipids is particularly

important because their reactivity and physical surface properties contribute to the formation and stabilization of the sea surface microlayer (SML) (Frka et al. (2012), and references therein). The SML represents a chemically distinct film enriched with surface active OM that accumulates at the air-water interface relative to the underlying water (ULW) (Wurl and Holmes, 2008;Wurl et al., 2011;Cunliffe et al., 2013). Furthermore, the SML, as a natural interface between ocean and atmosphere, could play an important role as OM source for aerosol particles in the marine environment (Engel et al., 2017). The presence

of lipids at the air-water interface is the result of their high surface affinity, competitive adsorption and segregation from other OM constituents (Frka et al., 2012).

OM can be transferred from the ocean to the atmosphere by wind-driven processes and bubble bursting, resulting in the formation of primary marine aerosol particles. Within this process, film drops leading to the formation of submicron particles are enriched mainly with OM compared to larger jet drops (Wilson et al., 2015). Long chain fatty acids and cholesterol as

constituents of aerosol particles are regarded as important factors for the activation of aerosol particles to cloud condensation nuclei (CCN) (Barati et al., 2019) or ice nucleation particles (INP) (Nguyen et al., 2017;DeMott et al., 2018). The presence of the TG lipid class serves as an indication that the aerosol particles consist to a certain extent of freshly emitted sea spray. Schiffer et al. (2018) reported that TG was broken down into fatty acids via the enzymes of triacylglycerol lipase and that these lipases therefore had the potential to alter the composition of sea spray aerosol. In laboratory studies by the authors Schiffer et

al. (2018), lipase enzymes have shown to be transferred from the ocean into the atmosphere. Until now, lipids or individual lipid classes have been often analyzed only in one compartment of the ambient marine environment, i.e. seawater or marine aerosol particles (Gagosian et al., 1982;Simoneit and Mazurek, 1982;Frka et al., 2011;Gašparović et al., 2014). The possible transfer of total lipids (determined as a sum parameter) has so far only been described under controlled conditions in laboratory studies (Rastelli et al., 2017) or for certain lipid classes such as fatty acids, *n*-alkanes, total hydrocarbons (Marty et al., 1979)



or as fatty acids and their derivatives (Cochran et al., 2016b) in freshly emitted sea spray aerosol. Unfortunately, comprehensive studies on marine lipid biogeochemistry and a possible transfer of certain lipid classes into the atmosphere are missing.

The present work aimed at investigating lipids at the Cape Verde Atmospheric Observatory (CVAO) as species representative for different lipid classes in the marine environment of the tropical Atlantic Ocean, to study their abundance, (biogenic) sources and selective transfer into the marine atmosphere. The lipid data set obtained for samples from different marine compartments at the CVAO is discussed in terms of biological and physical (INP) parameters. Finally, the potential transfer of the lipids from seawater to aerosol particles will be investigated by relating the physico-chemical properties of individual lipid classes to their respective observed atmospheric enrichment.

## 2. Experimental

### 2.1 Study area and sampling sites

As part of the MarParCloud project (Marine biological production, organic aerosol particles and marine clouds: a Process chain) with a contribution from MARSU (MARine atmospheric Science Unravelled: Analytical and mass spectrometric techniques development and application), a field campaign was performed at the Cape Verde Atmospheric Observatory (CVAO) from 13 September to 13 October 2017 (Triesch et al., 2020;van Pinxteren et al., 2019a). The CVAO, a remote marine station in the tropical Atlantic Ocean located on the northeast coast of Sao Vicente island (16º 51' 49" N. 24º 52' 02" S), is described in more detail in Carpenter et al. (2010) and Fomba et al. (2014). Concerted sampling was carried out during the campaign, including $PM_1$ aerosol particles at the CVAO and seawater at the ocean site (~16°53'30'N, ~24°54'00''W). The seawater sampling site was located upwind of the CVAO and had minimal island influence.

### 2.1.1 SML and seawater sampling

The SML samples (n=6) were collected using the manual glass plate technique (Cunliffe, 2014). A glass plate (500x250 mm) with a sampling area of 2500 cm$^2$ was briefly immersed vertically in the seawater and then slowly pulled upwards. The surface film material adsorbed on the surface of the glass plate was then moved directly into pre-cleaned bottles using Teflon wipers. The ULW samples (n=13) were collected from a depth of 1 m in pre-cleaned plastic bottles, which were attached to a telescopic rod. The bottles were opened under water at the intended sampling depth in order to avoid influences by the SML. In addition, 5-6 litres of bulk surface water (at a depth between 10 and 50 cm) were collected with a plastic bottle for pigment analysis (n=11).

### 2.1.2 Aerosol particles sampling

At the top of a 30 m high sampling tower at the CVAO, $PM_1$ aerosol particles (n=13) were investigated with a high-volume Digitel sampler DHA-80 (Walter Riemer Messtechnik, Germany) on 150 mm quartz fibre filters (Munktell, MK 360) at a flow rate of 700 L min$^{-1}$ in 24 h. The filters used for sampling were preheated at 105 °C for 24 h to avoid contamination. The collected filters were stored in aluminium boxes at -20 °C up until analysis.



## 2.2 Analytics

### 2.2.1    Seawater samples analytics: lipids, pigments, nutrients, microorganisms, INP activity

For lipid analysis, the seawater samples (2 L) were passed through a 200 µm stainless steel screen to remove zooplankton and larger particles. Afterwards, they were filtered through pre-combusted (350 °C for 5 h) 0.7 µm GF/F filters (Whatman®, Sigma

Aldrich). After the filtration step, the internal standard 2-hexadecanone (purity ≥ 98 %, Sigma Aldrich) (10 µg) was added to each filtrate, while the corresponding filters were stored in tubes at -78 °C until analysis. Dissolved lipids (DL) were immediately extracted from the filtrate with dichloromethane (DCM; dichloromethane for liquid chromatography LiChrosolv®, Merck) in four extraction steps (twice at pH 8 and twice at pH 2). The final extract was concentrated using a rotary evaporator. Particulate lipids (PL) were extracted by a modified one-phase solvent mixture of DCM-methanol-water

(Bligh and Dyer, 1959) after the addition of 2-hexadecanone as internal standard . DL and PL classes were further analyzed by Iatroscan thin layer chromatography-flame ionization detection (TLC- FID) (Iatroscan MK-VI, Iatron, Japan) as described in Gašparović et al. (2015) and (2017). After separation on Chromarod-SIII thin layer rods, lipid classes were identified and quantified by external calibration with a standard lipid mixture. The lipid classes investigated included HC, ME, FFA, ALC, 1,3 DG, 1,2 DG, MG, WE, TG, PP (including PG, PE, PC), GL (including SQDG, MGDG, DGDG), ST and PIG. Each sample

was measured in duplicate with a relative standard deviation < 10 %. Also, blank seawater samples (high purity water filled in pre-cleaned plastic bottles and handled the same as the seawater samples) were analysed and the blank values were always below 15 % of ambient seawater samples. All presented lipid values for SML and ULW samples were blank corrected by subtracting the field blank values from the samples.

For the pigment analysis, 5-6 L of bulk water were filtrated through GF/F filters. These were extracted in 5 mL ethanol, an

aliquot (20 µL) was injected into a high performance liquid chromatograph system (HPLC) with fluorescence detection (Dionex, Sunnyvale, CA. USA). The pigments, including chl-*a*, chl-*b*, fucoxanthin, pheophorbide, pheophytin a/b, zeaxanthin, diadinoxanthin, lutein, chlorophyllide, violaxanthin, β-carotene, were separated under gradient elution using methanol/acetonitrile/water systems as mobile phase, as described in van Pinxteren et al. (2019b). Nutrients covering nitrogen oxides ($N_2O_3$, $NO_2$, $NO_3$), phosphate ($PO_4^{3-}$) and silicates ($SiO_4^{4-}$) were measured colorimetrically according to Grasshoff et

al. (1999) with a Seal Analytical QuAAtro constant flow analyzer. Following Robinson et al. (2019), the prokaryotes were detected by characteristic signature in a side scatter plot and presented as total prokaryotic cell numbers (TCN). In order to distinguish between prokaryotic and eukaryotic autotrophs, the approach described in van Pinxteren et al. (2019a) was used to study *Synechococcus-like cells* and *Nanoeucaryotes*. Two droplet freezing devices called LINA (Leipzig Ice Nucleation Array) and INDA (Ice Nucleation Droplet Array) were used at TROPOS to obtain information on the ice nucleation activity as

described in more detail by Gong et al. (2020), and references therein.

### 2.2.2    Aerosol particle samples analytics

The analysis of sodium ($Na^+$) was performed by ion chromatography (ICS3000, Dionex, Sunnyvale, CA, USA) as described in Mueller et al. (2010). For the analysis of lipid classes, 28.27 cm$^2$ of the PM$_1$ samples were extracted and measured following





the procedure for particulate lipids in seawater (see section 2.2.1). A chromatogram of the TLC-FID measurements of an aerosol particle sample and the standards is shown in Fig. S1. The field blanks (n=5) were prepared using pre-baked quartz fibre filters without active sampling and treated according to same procedure as the field samples. The concentrations of the lipid classes were calculated by external calibration. Each sample was measured in duplicate with a relative standard deviation

< 10 % and field blanks, which were always below 20 % of real aerosol particle sample, were subtracted. All presented values are blank corrected.

### 2.2.3    Lipolysis Index

The Lipolysis Index (LI) is an index describing the lipid degradative state and the biological degradation processes of lipids

(Goutx et al., 2003). In this concept, it is proposed that the degradation process of marine acyl-lipids by the concentration ratio of free lipids/metabolites (ALC, FFA, MG, DG) to their precursors (TG, WE, glycolipids as MGDG, DGDG, SQDG and phospholipids as PG, PE, PC) could be described by equation (1).

$$LI = \frac{(ALC+FFA+MG+DG)}{(TG+WE+MGDG+DGDG+SQDG+PG+PE+PC)} \qquad (1)$$

It is obvious that the concentration of precursor lipids and metabolites alone can already influence the LI and that there may be a natural variance between them. For example, metabolites can be quickly absorbed by existing microorganisms, which would lead to a reduction of the concentration and thus influence the LI. However, the LI is reported as a useful measure to characterize the degradation level of biogenic organic material (Goutx et al., 2003). Higher LI values are characteristic for

enhanced OM degradation and metabolite release, while lower LI values indicate that the appearing lipid classes are very fresh or resistant to degradation.

### 2.2.4    Enrichment factors

The enrichment factor in the SML ($EF_{SML}$) was calculated by dividing the concentration of the respective analyte in the SML

by the concentration of the analyte in the ULW using equation (2):

$$EF_{SML} = \frac{c\,(analyte)_{SML}}{c(analyte)_{ULW}} \qquad (2)$$

An enrichment in the SML is defined as $EF_{SML} > 1$, a depletion with $EF_{SML} < 1$.


To calculate the enrichment factor of the different analytes in aerosol particles ($EF_{aer}$) relative to the SML, the atmospheric concentration of the analyte relative to the sodium concentration on the $PM_1$ sample was divided by the analyte concentration relative to the sodium concentration in the corresponding SML sample using equation (3):

$$EF_{aer} = \frac{c\,(analyte)_{PM_1}/c\,(Na^+)_{PM_1}}{c\,(analyte)_{SML}/c\,(Na^+)_{SML}} \qquad (3)$$





The EF$_{aer}$ calculation was limited by the availability of the analyte concentration in both matrices, i.e. PM$_1$ and SML samples collected simultaneously.

### 2.2.5 Statistical analysis

5  To investigate possible relationships between chemical, (micro)biological and physical parameters of the seawater samples (ULW, SML), listed in Fig. S2, a Pearson correlation analysis was performed. Figs. S3-S6 show the matrix of parameters when either ULW or SML samples of the dissolved or particulate fraction are considered. The correlation coefficient (R), number of samples examined (n) and the p-value were used to validate the significance of the correlation. In particular, the p-value as a test for statistical hypothesis in research areas must be considered when defining statistical relevance (Bhattacharya and

10 Habtzghi, 2002;Perezgonzalez, 2015). The statistical parameters for the performed analysis of the ULW and SML samples are defined as follows. For the ULW samples, the statistical relevance of a relationship was defined if $\geq 4$ (n$_{max}$=13), R $\geq$ 0.6 and p-value $\leq$ 0.05 (Perezgonzalez, 2015). We considered a 'trend' to be valid if n $\geq$ 4 (n$_{max}$=13), R $\geq$ 0.4 and p-value $\geq$ 0.05. Due to the limited number of SML samples, p-values were always $\geq$ 0.05, so we could not define statistical relevance, but considered a 'trend' to be valid if n $\geq$ 4 (n$_{max}$=6), R $\geq$ 0.4.

### 3. Results and Discussion

### 3.1 Seawater and SML samples

### 3.1.1 Lipids and lipid classes in the particulate fraction

The particulate lipids showed a certain variance during the campaign, as shown in Fig. 1. For the ULW samples, the

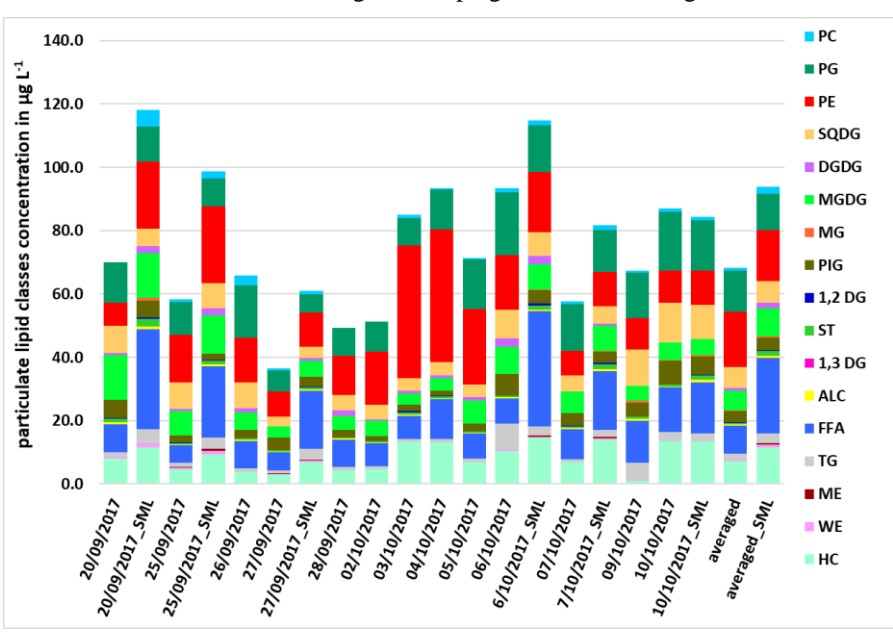

**Figure 1: Concentration of individual particulate lipid classes in the ULW and the SML samples along the campaign and as an averaged value in µg L$^{-1}$**





concentration of ∑PL was between 36.4 and 93.5 µg L⁻¹, for the SML samples between 61.0 and 118.1 µg L⁻¹. The measured total lipid concentrations are within the concentration range of the previous measurement studies in different oceanic regions (Frka et al., 2011;Gašparović et al., 2014;Stolle et al., 2019). The lipid classes FFA and phospholipids (PE, PG, PC) dominated within the PL while other lipid classes like TG, HC, ST showed not only a much lower concentration but also much less

pronounced variance. Another interesting feature is the opposite abundance of the two phospholipids PE and PG. On sampling days when PE had a high concentration, PG was significantly less concentrated as found in the middle of the campaign (e.g. 3/10/2017 and 4/10/2017, see Fig. 1), whereas towards the end of the campaign (e.g. 07/10/2017 and 09/10/2017), when the concentration of PE becomes lower, the concentration of PG increased. Via the PE/PG ratio the contribution of bacterial lipids to the OM pool in seawater can be retrieved (Goutx et al., 1993). The PE/PG ratio varied along the campaign with increasing

values towards the middle of the campaign (maxima on 03/10/2017 and 04/10/2017) and decreasing values afterwards (Table S4), following the same trend as the total bacteria number (TCN, Table S3). This indicates a change in the lipid dominant biological contributions, with bacterial sources dominating in the first part and especially in the middle of the campaign, whereas in the last part (afterwards on 05/10/2017) rather phytoplankton-dominated contributions to the lipid pool. These differences between bacterial and phytoplankton sources are not reflected in the total lipid concentrations, because degradation

products like FFA also contribute strongly to the total lipids (Fig. 1, S7). For this reason, neither bacterial nor phytoplankton sources alone are the controlling drivers for the total lipid concentration, at least in the short term.

**3.1.2 Lipids and lipid classes in the dissolved fraction**

Compared to the particulate fraction, slightly higher concentrations of total dissolved lipids were detected, 39.8-128.5 µg L⁻¹

in the ULW and 55.7-121.5 µg L⁻¹ in the SML samples (Fig. 2). The concentrations reported here were slightly higher than the total dissolved lipid concentrations reported by Frka et al. (2011) in the Mediterranean semi-enclosed temperate Adriatic sea (∑DL: 7.5-92.2 µg L⁻¹).

In contrast to the particulate lipids, HC showed the highest concentration and variation within the lipids and varied between 6.6 up to 64.0 µg L⁻¹ in the ULW and in the SML between 9.2-49.6 µg L⁻¹ (Fig. 2). HC can have both anthropogenic and natural

sources (Scholz-Böttcher et al., 2009). Shorter *n*-alkanes (as 2-nonadecane) may have additional sources as mature organic matter or petroleum contamination, but the main sources of the investigated HC surrogate (2-nonadecane) is marine phytoplankton (Scholz-Böttcher et al. (2009) and references therein).

Phospholipids, especially PE and PG, as well as FFA, which dominated in the particulate lipids, showed significantly lower concentrations within the total dissolved lipids. In contrast to the particulate fraction, both phospholipids (PE and PG) showed

most similar concentrations and percentages in the dissolved fraction.





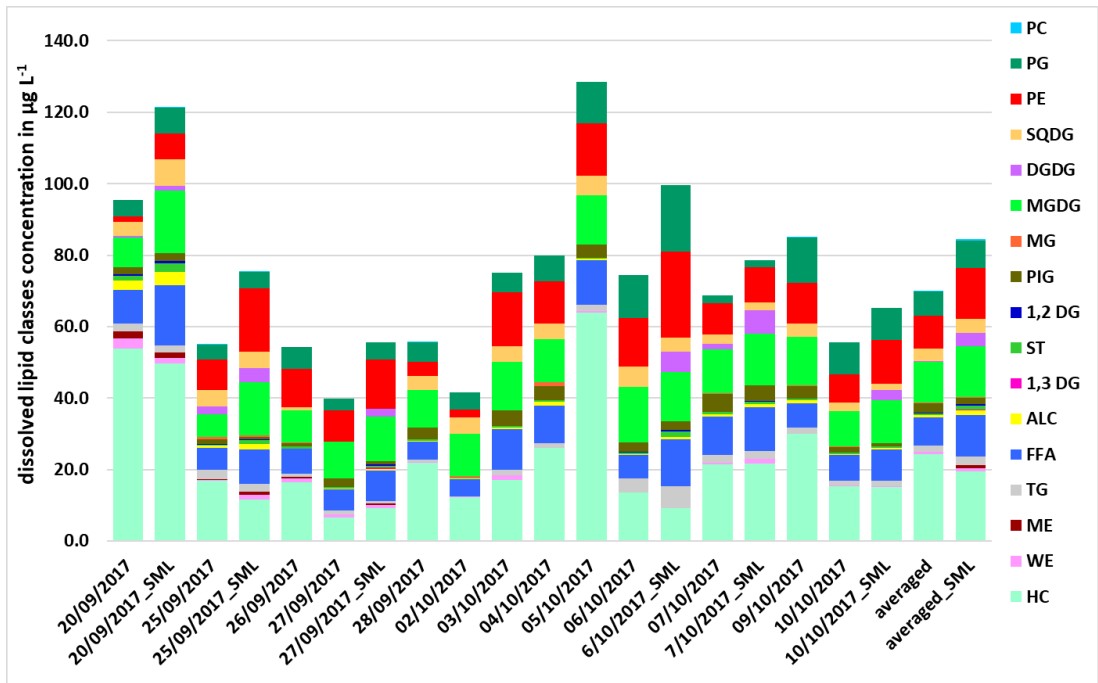

**Figure 2: Concentration of individual dissolved lipid classes in the ULW and the SML samples along the campaign and as an averaged value in µg L$^{-1}$**

The percentage composition of total dissolved lipids (Fig. S8) is in good agreement with the literature (Goutx et al., 2009;Marić

et al., 2013). Altogether, our detailed analysis of the lipid classes shows that, although the concentration of PL and DL are overall very similar, the composition of the lipids to the PL and DL groups is partly different. This indicates that different production and degradation mechanisms for DL and PL contribute to the respective lipid composition.

### 3.1.3 Lipolysis Index

To evaluate the lipid degradative state and (bio)degradation processing, the LI (Eq. (1); Table S5) was calculated for the lipids in the seawater samples. The LI in the particulate fraction varied between 0.13-0.31 in the ULW and between 0.37-0.66 in the SML samples. Van Wambeke et al. (2001) reported LI values of particulate lipids (0.21-0.39) in the north-western Mediterranean Sea during the end of a phytoplankton bloom up to pre-oligotrophic conditions. In general, the LI indicated that the intact lipid classes were dominant compared to the degradation indices/metabolites for the particulate lipids and that the

lipids were thus degraded only to a small extent. This coincides well with the low concentrations of chlorophyll degradation products, suggesting that only moderate grazing took place and the (pigment-containing) organisms were fresh and in healthy condition (van Pinxteren et al., 2019a). However, on specific days, the LI$_{SML}$ of PL was ≥ 0.5 (Table S5), indicating increased OM/lipid degradation and metabolite release in SML compared to ULW. This observation was also made by Gašparović et al. (2014) in the East Atlantic Ocean and can be attributed to both bacterial and photochemical abiotic degradation (Christodoulou





et al., 2009). The LI of DL (Table S5) varied between 0.13-0.53 in the ULW and between 0.20-0.48 in the SML samples, suggesting that the dissolved lipid classes were quite resistant to degradation.

### 3.1.4 Pigments, nutrients and microbiological investigations in seawater

To further elucidate the biological production and degradation state of lipids, lipid concentrations were related to a set of biological parameters, including indicators for autotrophic organisms (namely marine pigments, chl-*a*, *Nanoeucaryotes* and *Synechococcus-like cells)* and TCN as a proxy for bacterial abundance. Altogether, the concentrations of pigments and autotrophic and heterotrophic cells indicated an oligotrophic system (detailed values in Tables S2/S3, further information is given in van Pinxteren et al. (2019a)).

The pigment measurements of the bulk water indicated temporal changes in the composition of the community and an increasing trend in pigment concentration towards the end of the campaign. A correlation of total lipids with the abundances of chl-*a, Nanoeucaryotes and Synechococcus-like* cells was not observed. However, with regard to specific pigments beyond chl-*a*, a statistically relevant correlation between particulate PE and the pigment zeaxanthin with R=0.69 (p-value=0.03, n=10) was found in the ULW (Fig. S9a). Zeaxanthin has been reported as a proxy for chlorophytes and cyanobacteria (Grant and
Louda, 2010) and for some microalgae (Galasso et al. (2017), and references therein). Furthermore, the pigment fucoxanthin, a marker for diatoms (Descy et al., 2009), showed a weak trend with particulate FFA (R=0.53, p-value=0.12, n=10) in ULW samples (Fig. S9b)**.** The observed correlations/trends of zeaxanthin and fucoxanthin with individual particulate lipid classes pointed to a contribution of chlorophytes, cyanobacteria and diatoms to the lipid pool in our study.

Regarding the heterotrophic parameters, a positive and statistically relevant correlation between PE and TCN was observed in
the dissolved fraction of ULW samples (R=0.79, p-value=0.006, n=10, Fig. S10a). We also found a positive correlation between the particulate PE and TCN in the ULW (R=0.72, p-value=0.02, n=10), a similar trend was noticed for PE and TCN in the SML (R=0.64, p-value=0.36, n=4) (Fig. S10b). The contribution of bacteria to the PE pool most likely results from the fact, that PE is part of the bacterial membrane (Stillwell, 2016;Gašparović et al., 2018). Additional indications for bacteria influencing the lipid pool results from a negative correlation between the lipolysis index of total particulate lipids ($LI_{PL}$) and
TCN in ULW and SML samples. In the ULW, a correlation between $LI_{PL}$ and TCN with R= -0.73 (p-value=0.02, n=10) (Figure S10c) was observed. In the SML (Figure S10c), a similar trend was noticeable for $LI_{PL}$ and TCN (R= -0.87, p-value=0.13, n=4). These relationships may result from the passive contribution (higher cell abundance and thus higher concentration of PE) of TCN to the phospholipids pool via PE. Phospholipids contribute to LI as part of the intact lipids (section 2.2.3, Eq. 1), which leads to a negative correlation with $LI_{PL}$. On the other hand, the 'metabolites' could be actively taken up by bacteria,
which most likely happens when more bacteria are present. Although it remains unclear whether the bacteria have a passive (i.e. via membrane) or active (i.e. metabolism of the lipid 'metabolites') effect on the observed correlation between $LI_{PL}$ and TCN, it is most likely that the bacteria have influenced the lipid pool which is consistent with the results obtained from the lipid composition. In addition, the results underline that chl-*a*, as commonly used proxy for bioproduction, may not sufficiently explain the variability of the lipid classes in the marine region studied.



### 3.2 Transfer of lipids from the Oceans

### 3.2.1 Enrichment in the SML

The $EF_{SML}$ was calculated using Eq. (2) to compare the concentration of the lipid classes in the SML samples with the ULW samples. The $EF_{SML}$ of the total lipids and of lipid class representatives in the particulate and dissolved fraction is listed in

Table S6. For the total lipids in the particulate fraction, the $EF_{SML}$ varied between 1.0-1.7 (averaged $EF_{SML(\sum PL)}$: 1.4), whereas in the dissolved fraction it varied between 1.1-1.4 (averaged $EF_{SML(\sum DL)}$: 1.3). The $EF_{SML}$ of the total lipids are therefore quite similar for PL and DL. The slightly higher enrichment of the particulate fraction compared to the dissolved fraction in the SML is in good agreement with the literature (Marty et al., 1988;Gašparović et al., 1998;Kuznetsova and Lee, 2002;Kuznetsova et al., 2005;Burrows et al., 2014). The preferred enrichment of the particulate fraction is probably due to the fact that the

bubbles capture larger particles more efficiently because of their larger radius and inertia (Sutherland, 1948;Weber et al., 1983;Dai et al., 1998) and that the gelatinous nature of the SML can ensure that the particulate OM is captured (Robinson et al., 2019). Moreover, marine dissolved lipids can be produced by dissolution from the particulate fraction and through primary production and released during the life cycle and after cell death (Yoshimura et al., 2009;Novak et al., 2018). They might lead to a slightly higher SML enrichment of the particulate lipids.

A major aspect contributing to lipid enrichment might be explained by the physico-chemical properties of the respective lipid classes, namely the surface activity (Table S7). The parameters describing this characteristic, i.e. the density, the partitioning coefficient between octanol and water ($K_{OW}$) and the topological polar surface area (TPSA), were compared with lipid enrichment. As shown in Table S7, the nonpolar lipids such as FFA and ALC have a higher surface accumulation potential compared to the more polar glycolipids and phospholipids, but a correlation between enrichment of lipids and surface activity

was absent. In the dissolved fraction neither such a gradation of enrichments at the surface, nor a correlation between enrichment and parameters describing surface activity were visible. As Table S4 shows, similar $EF_{SML}$ were found in the dissolved fraction for the individual lipid classes ($EF_{SML}$: 1.5 (FFA), 1.7 (ALC) and 1.6 (PP)). A comparison of lipid enrichment with other OM compound groups showed that SML enrichment of lipids seemed to be less pronounced in contrast to other organic species such as amino acids (Reinthaler et al., 2008;Triesch et al., 2020) This is somehow surprising, since SML

enrichment of lipids is likely to be strongest among organic groups due to their surface activity (Burrows et al. (2014), and references therein). It has to be considered that a SML described here represents a layer with a thickness of about 100 µm (van Pinxteren et al., 2017) and therefore gradients within this layer (e.g. an enhanced enrichment of surfactants only in the top layer of a few µm) cannot be regarded here. The fact, that other (less surface active) compounds are stronger enriched in the SML (upper 100 µm) underlines the need to consider additional parameters to describe the SML enrichment of lipids in the

ambient marine environment.

To this end, besides the physical processes leading to SML enrichment, the in-situ production of OM (bacteria, phytoplankton and their released metabolites) were further investigated. Regarding the enrichment in the SML within the lipid classes or within both fractions (dissolved and particulate), stronger differences are found when looking at the individual lipid classes. The bacterial marker PE was enriched in the SML in DL and PL fractions with $EF_{SML(PE)}$ of 1.6 (PF) and with 2.1 (DF). In





contrast, the phytoplankton marker PG was always depleted in the SML ($EF_{SML} < 1$) in the PL and mostly enriched in the DL (averaged $EF_{SML(PG)}$ of 1.3). This is consistent with the observed permanent abundance and slight enrichment of TCN and indicates enhanced bacterial activity in SML. Furthermore, the degradation lipid class FFA showed high enrichments in the SML (averaged $EF_{SML(FFA)}$ of 3.1 (PL) and 1.5 (DL)). These high concentrations and enrichments point to an enhanced

biodegradation in the SML, which is consistent with previous observations, that lipids are degraded more strongly in the SML (high LI) than in the ULW. This may be due to a different diversity and different taxa of microorganisms in the SML that can differ significantly from those in the underlying water (Cunliffe et al., 2011). The metabolic reserves lipids, represented by TG, showed the highest variability of enrichment in the SML along the campaign in the particulate fraction. In the particulate fraction, $EF_{SML(TG)}$ varied between 0.3 and 4.4, resulting in an averaged enrichment of 2.3. The enrichment in the lipid classes

of the dissolved fraction was less pronounced and always showed an opposite trend to PL, i.e. if TG was highly enriched in PL, it was less enriched in DL and vice versa. This indicates that the lipid reserves are stored in the particulate lipids and are dissolved producing dissolved TG.

Altogether, our results indicate that physical processes alone, which are related to the surface activity of the lipids, are not sufficient to describe the SML enrichment of the lipids, at least not in the top 100 µm. In-situ formation and degradation by

phytoplankton and mainly bacterial processes, as shown here from the lipid classes patterns, also contribute to the abundance and SML enrichment of lipids in the ambient marine seawater.

### 3.2.2 Measured PM$_1$ aerosol particle composition

Up to now, the discussion about lipids on (marine) aerosol particles has only covered distinct classes of lipids (mostly fatty

acids). Given this fact, this work firstly presents a comprehensive analysis of several lipid classes on marine aerosol particles. The atmospheric concentration of total lipids in PM$_1$ samples at the CVAO varied between 75.2 and 219.5 ng m$^{-3}$ (average 119.9 ng m$^{-3}$), as shown in Fig. 3. The atmospheric concentration at the CVAO was 18.5 ng m$^{-3}$ (8.7-33.9 ng m$^{-3}$) for FFA and 6.3 ng m$^{-3}$ (3.4-9.8 ng m$^{-3}$) for ALC. This was in good agreement with Kawamura et al. (2003), who reported atmospheric concentrations between 0.19-23 ng m$^{-3}$ (average 2.0 ng m$^{-3}$) for ALC and between 2.5-38 ng m$^{-3}$ (average 14 ng m$^{-3}$) for FAA

on marine aerosol particles from the western North Pacific. Other than that, Mochida et al. (2002) observed atmospheric concentration between 0.8-24 ng m$^{-3}$ for saturated fatty acids (C$_{14}$-C$_{19}$) on marine aerosol particles collected overt the northern Pacific.

The percentage contribution of the individual lipid classes to the total lipids is shown in Fig. S11. A high percentage contribution of the lipid classes TG, FFA and ALC (26.3-64.0 %, average 39.8 %) to the total lipids was observed. Especially,

high percentages for TG (11.9-29.1 %, average 18.8 %) and FFA (8.4-31.2 %, average 15.4 %) were noticeable (Fig. S10). Compared to the seawater lipids, the atmospheric composition showed the same classes of lipids with a stronger agreement to the DL composition (high contribution of HC and lower contributions of phospholipids).

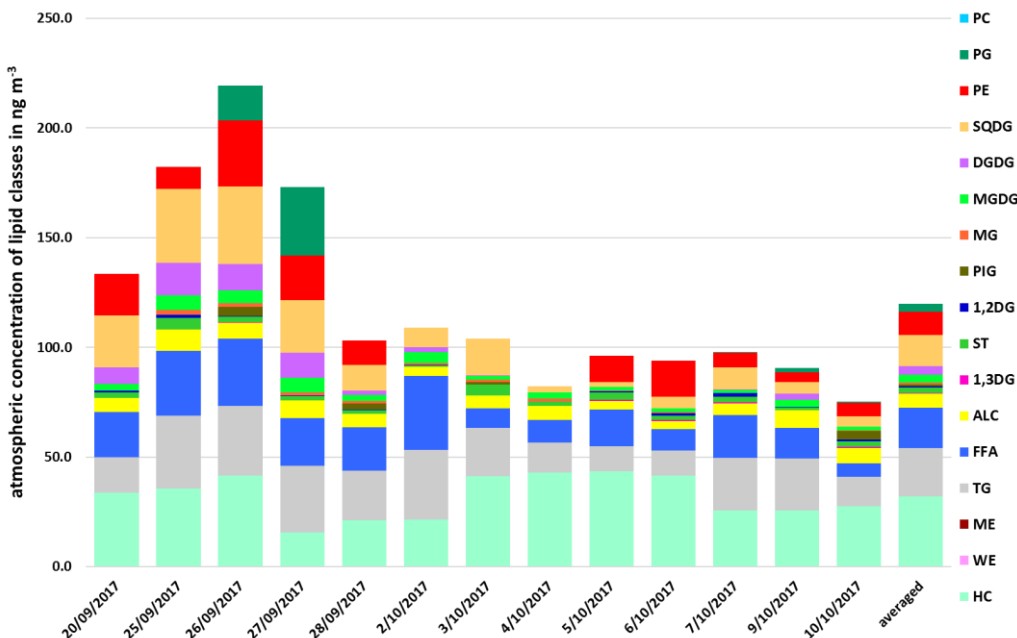

**Figure 3: Atmospheric concentration of individual lipid classes in PM$_1$ aerosol particle samples and as an average at the CVAO in ng m$^{-3}$**

However, although the atmospheric concentration of phospholipids was lower, it was found that PE was always higher concentrated as PG, with one exception on 27/09/2017 (Fig. 3). Since heterotrophic bacteria are reported as a dominant sources of PE (Michaud et al., 2018), this suggests that i) bacteria, possibly transported from the ocean into the atmosphere, produce PE on the aerosol particles and/ or ii) PE is directly transferred from the ocean into the atmosphere, likely via bubble bursting. The high presence of TG on the aerosol particles (Fig. 3) strongly suggests that the aerosol particles consist to a certain extent of freshly emitted sea spray. This is consistent with the observation that the submicron aerosol particle samples at the CVAO were mainly maritime influenced during this period, based on particulate mass, and showed only minor dust or anthropogenic impacts (Triesch et al, 2020).

### 3.2.3 Transfer of lipid classes from the ocean to the aerosol particles

An often applied parameter to quantify the transfer of OM from the ocean to the atmosphere, is the EF$_{aer}$ (e.g. Russell et al. (2010), van Pinxteren et al. (2017), Triesch et al. (2020)). According to Eq. (3), the EF$_{aer(TL)}$ was calculated based on the dissolved total lipids in SML and varied between $9 \cdot 10^4$ and $7 \cdot 10^5$ with an average value of $3 \cdot 10^5$ (Fig. 4, Table S8). The data reported in the literature for enrichment factors of organic carbon or groups of OM in aerosol particles often originate from laboratory experiments, e.g. using controlled artificial bubbling unit (Quinn et al. (2015), and references therein). Rastelli et al. (2017) determined the enrichment of lipids, as a sum parameter, on submicron aerosol particles compared to seawater in a bubble-bursting experimental set-up and found an EF$_{aer}$ of $1 \cdot 10^5$. The good agreement of the EF$_{aer(TL)}$ derived from the ambient





measurements reported here with the $EF_{aer}$ derived from laboratory experiments under controlled conditions by Rastelli et al. (2017) indicates that the transfer of lipids from the SML to the aerosol phase under ambient conditions is consistent with processes described in laboratory studies.

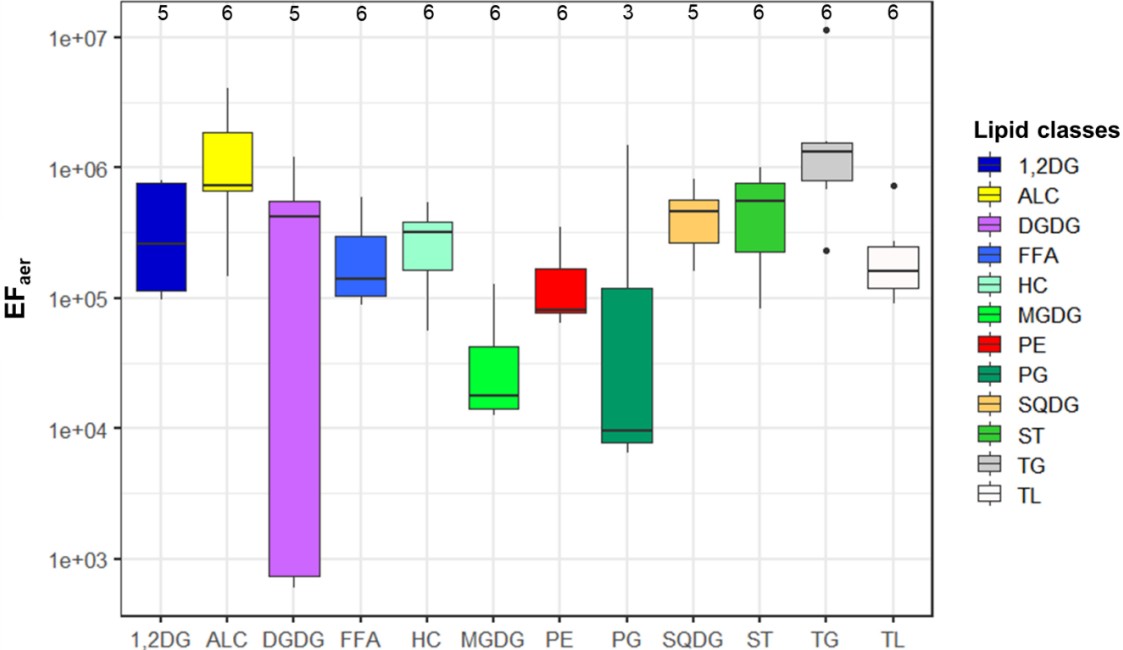

**Figure 4: Boxplot of the enrichment factor aerosol ($EF_{aer}$) of the individual lipid classes and total lipids (TL) at the CVAO including the median, the 25th and 75th percentile; more explanation in Fig. S12**

The $EF_{aer}$ of total lipids was about one to two orders of magnitude higher than the $EF_{aer}$ of free amino acids ($4 \cdot 10^2$-$3 \cdot 10^4$) on submicron aerosol particles measured during the same campaign (Triesch et al., 2020), underlining the preferred transfer of lipids from the ocean to the atmosphere. In contrast to SML enrichment, the higher enrichment of lipids on aerosol particles

observed here corresponds well with the high surface activity of the lipids and the preferred adsorption to (bubble) surfaces resulting in a strong sea-to-air-transfer (Tervahattu et al., 2002;Facchini et al., 2008;Cochran et al., 2016a;Schmitt-Kopplin et al., 2012;Rastelli et al., 2017), and their possible association with other compounds promoting co-aerosolization processes (Quinn et al., 2015;Hoffman and Duce, 1976;Rastelli et al., 2017). Further possible transport mechanisms are discussed in section 3.2.4.

A gradient regarding $EF_{aer}$ of the individual lipid classes was found, showing that some of them were enriched to a larger extent than others (Fig.4, Table S9). Such differences between the lipid classes have not been reported so far because the lipids were mainly measured as a sum (Rastelli et al., 2017) or only a specific lipid class (as FFA, HC) has been investigated (Cochran et al., 2016b;Marty et al., 1979). In our data set we observed that TG ($EF_{aer(TG)}$: $3 \cdot 10^6$) followed by ALC were most enriched ($EF_{aer(ALC)}$: $1 \cdot 10^6$), while MGDG showed a lower enrichment with $EF_{aer(MGDG)}$: $4 \cdot 10^4$. For the SML enrichment, the parameters

describing the surface activity (density, $K_{OW}$ and TPSA, Table S7), were compared with lipid enrichment. Lipid classes with



comparatively low surfactant activity (in relation to the TPSA value of the lipid class, Table S7) including PP and GL showed relatively lower enrichments ($EF_{aer(PP)}$: $2 \cdot 10^5$ and $EF_{aer(GL)}$: $3 \cdot 10^5$) compared to highly enriched ALC ($EF_{aer(ALC)}$: $1 \cdot 10^6$ ). Furthermore, a mild connection ($R^2$=0.43, p=0.028) was found between the log $K_{OW}$ and the $EF_{aer}$ of the individual lipid classes (Fig. S13), indicating that the compounds with higher log $K_{OW}$ and therefore stronger lipophilicity are preferably enriched on

the aerosol particles. For example, TG with the highest log $K_{OW}$ value of 25.5 (Table S7) was observed to have the highest enrichment on aerosol particles ($EF_{aer(TG)}$: $3 \cdot 10^6$). In contrast, lipid classes with lower log $K_{OW}$ values such as MGDG (log $K_{OW(MGDG)}$: -3.5) were characterized by lower enrichments ($EF_{aer(MGDG)}$: $4 \cdot 10^4$). The compounds that are highly enriched in the aerosol phase only partially correspond to their respective enrichment in the SML. TG and ALC showed a high enrichment both in the SML and in the particle phase. However, FFA, that showed a pronounced SML enrichment in PL ($EF_{SML(FFA)}$:3.1),

exhibited only a medium enrichment in the aerosol particles compared to other lipid groups.

It needs to be emphasized, that the calculated $EF_{aer}$ provides the quantitative description of the transfer from the ocean to the atmosphere, but does not consider additional formation or degradation pathways of lipids on the aerosol particles, including biological or photochemical atmospheric reactions and a transport from other than marine sources. In agreement with the results of SML enrichment, these results suggest that additional processes such as biotic formation and degradation influence

the lipid abundance on the aerosol particles. It has been reported that microorganisms (Rastelli et al., 2017) and especially bacteria (Michaud et al., 2018) can be transported from the ocean into the atmosphere. Bacteria can be transferred to marine aerosol particles and may produce or degrade lipids. In addition, photochemical oxidation processes can take place in the atmosphere, i.e. conversion of FFA to ALC (Bikkina et al., 2019).

Overall, the detailed measurements of the lipid classes within the concerted measurements together with additional parameters

showed that although lipids are the highly OM species in aerosols, which is in-line with their high surface activity, additional biological processes influence the lipid composition. These need to be further studied and considered in OM transfer models. More recent models recognized that OM transfer must be modeled by including the individual OM groups (like lipids) rather than phytoplankton tracers like chl-*a* (Burrows et al., 2014). Indeed, these models describe OM transfer on the basis of their physico-chemical properties, but our data suggest that in the ambient marine environment, additional in-situ formation and

degradation must also be considered in order to fully address OM abundance in general and the lipid abundance in particular.

### 3.2.4 Discussion of possible transfer mechanisms

The transfer of the dissolved and particulate OM from the ocean to the atmosphere probably occurs via bubble bursting, whereby bubbles rising through the water column absorb OM and, when bursting at the surface, release the OM to the aerosol

particles via jet and film drops (Quinn et al. (2015), and references therein). The finding here, that both DL and PL, contain similar classes of lipid, which are also found on the aerosol particles (Fig. 5, Fig. S14), suggests that both types of lipids in seawater are transferred to the aerosol particles via bubble bursting process. A differentiation of the contribution of the dissolved and particulate lipid fractions in seawater to the formation of the aerosol particles was therefore not possible here.





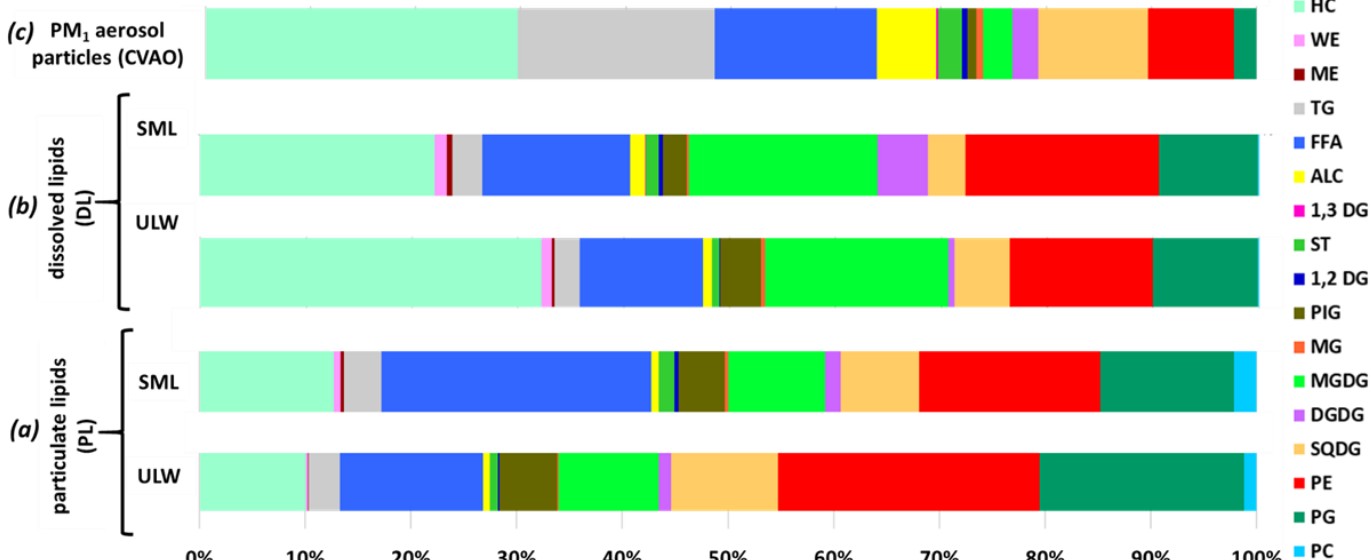

**Figure 5: The percentage contribution of the individual lipid classes to the total lipids in the (a) particulate and (b) dissolved fraction in seawater (differentiation between ULW and SML) and on (c) PM₁ aerosol particles at the CVAO**

However, regarding the process of OM absorption on the bubbles in more detail, the OM absorbed on the bubble can be

distributed either towards the gas or aqueous phase or can preferably reside within the bubble interface. To conceptually address the distribution of the OM towards the bubble-water-interface, we calculated the adsorption coefficient related to air ($K_a$) after Kelly et al. (2004) and additionally an adsorption coefficient of the analytes related to water ($K_{aq}$). The calculation included the measured SML concentrations, Henry's Law constants and the gas-phase saturation vapor pressure as explained in detail in Table S10. The comparison of the calculated $K_a$ and $K_{aq}$ values can give a hint on the distribution of lipids at the

bubble-air-water-interface. When $K_{aq} \gg K_a$ (Fig. S19a), the analyte should be preferred distributed (from water) to air (inside the bubble). When $K_a \gg K_{aq}$ (Fig. S19b) in turn, the analyte should be preferably distributed (from air) into water while the analyte should be preferred distributed within the bubble interface when $K_{aq} \sim K_a$ (Fig. S19c). Our data set showed that the analytes that had similar $K_a$ and $K_{aq}$ values (Table S10), namely TG and ALC, had highest $EF_{aer}$ ($EF_{aer(TG)}$: $3 \cdot 10^6$ and $EF_{aer(ALC)}$: $1 \cdot 10^6$). This indicates that analytes, which are preferably distributed within the interface of the bubble, are transferred to the

aerosol particles to a larger extent via the bubble bursting process, probably due to the higher stickiness to the interface (Fig. 6).



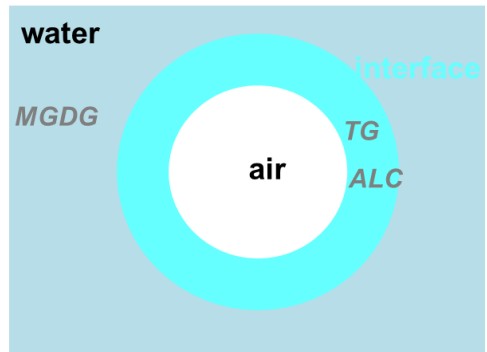

**Figure 6: Scheme of a bubble during the bubble bursting process, distinguished between 'air' (inside the bubble), 'water' (surrounding the bubble), the 'interface' (bubble surface) and the distribution of the lipid classes MGDG, TG and ALC related to their $K_a$ and $K_{aq}$ values**

For MGDG, the lipid class with the lowest $EF_{aer}$ ($4 \cdot 10^4$), the observed ratio was $K_a \gg K_{aq}$, meaning that MGDG was preferably distributed in water. For the other lipid classes, however, we did not find such a connection between the adsorption coefficients ($K_{aq}$, $K_a$) and the aerosol enrichment ($EF_{aer}$). Nevertheless, the hypothesis that the transfer and enrichment of the lipids is related to the distribution of a compounds within the bubble-air-water-interface, as observed for the lipid classes with extreme $EF_{aer}$, should be further investigated, preferably in controlled laboratory experiments.

### 3.3  Connection between lipids and INP activity in seawater

One main feature of biological components in general is their potential ability to contain ice nucleating abilities and act as INP in the atmosphere (Šantl-Temkiv et al. (2019) and references therein). To identify potential connections between lipid classes and INP activity in seawater, a statistical analysis was performed as described in section 2.2.5. As shown in detail in Gong et

al. (2020), all samples collected for the present study contained both SML and ULW INP with concentrations of ~ 200 $L^{-1}$ at temperatures of about -10 °C (Fig. S17), increasing to $10^7$ $L^{-1}$ at ~ -25 °C. The existence of INP that are already ice active at temperatures above -15 °C indicated the presence of biogenic INP (Kanji et al., 2017;Šantl-Temkiv et al., 2019).

Both lipid fractions of the SML samples showed a positive trend towards the INP concentrations measured at -10 °C (R=0.72, p-value=0.28, n=4 for the total PL and R=0.69, p-value=0.31, n=4 for the total DL, Fig. S15a/b). This connection suggests the

involvement of lipids in biogenic INP. Furthermore, the lipid classes which had shown a relationship with autotrophic or heterotrophic organisms, namely PE and FFA (section 3.1.4), were investigated for their INP relationship. The relationship between PE and INP with regard to ULW cannot be discussed here, as the criteria defined in section 2.2.5 were not met (e.g. not enough matching data points were available). But a trend was found between SML particulate PE and INP measurements at -10 °C (R=0.95, p-value=0.06, n=4, Fig. S15c). Similar relations between lipids and INP activity have been reported

previously (Govindarajan and Lindow, 1988;Palaiomylitou et al., 1998;DeMott et al., 2018). Palaiomylitou et al. (1998) reported that ice nucleation proteins were associated with phospholipids and showed that phospholipids, especially PE, not only contribute to increased overall activity but also to the production of ice nuclei active at higher temperatures. To further





test biogenic INP activity, we analyzed the INP activity of the seawater samples before and after heating (95°C for 1 hour), since biogenic, especially proteinogenic, compounds are deactivated when heated to 100 °C (Šantl-Temkiv et al., 2019). It could be shown that a large proportion of INPs, that were active between -10 °C and -15 °C, lost their ice activity after the heating procedure (Fig. S17). As mentioned above, the deactivation of the INP function by heating is often associated with

proteins. However, it has been shown that ice nucleating proteins have a connection to lipids and interactions with membrane lipids, especially PE, are needed to maintain the conformational structure and functional activity of many membrane-bound proteins (Govindarajan and Lindow, 1988;Palaiomylitou et al., 1998). For this reason, the lipids might have a driving function in the INP activity of biogenic INP. In the SML samples, trends between INP measurements at -10 °C and the particulate FFA (R=0.84, p-value=0.16, n=4, Fig. S16a) and dissolved FFA (R=0.63, p-value=0.37, n=4, Fig. S16b) were observed. Moreover,

a trend was found between the particulate FFA in the ULW and the INP measurements at -15 °C (R=0.64, p-value=0.025, n=12, Fig. S16c). DeMott et al. (2018) reported that ice nucleation by particles containing long-chain fatty acids in a crystalline phase was relevant for freezing by sea spray aerosols. Burrows et al. (2013) suggested that marine biogenic INP most probably played a dominant role in the INP concentrations studied in near-surface-air over the Southern Ocean. Wilson et al. (2015) proposed that there are ice active macromolecules in the OM of SML. Moreover, they pointed out that global model simulations

of marine organic aerosol in connection with their measurements indicated that marine OM might be an important source of INP in remote marine environments, e.g. the Southern Ocean, North Pacific Ocean and North Atlantic Ocean (Wilson et al. (2015), and references therein). The relationships presented here between the lipids in general and in particular the lipid classes with assigned biological context (PE, FFA) and INP activity at higher temperatures (-10 °C, -15 °C) in the ambient SML samples are consistent with the results of Wilson et al. (2015) indicating that lipids in the tropical North Atlantic Ocean have

the potential to contribute to (biogenic) INP activity when transferred to the atmosphere. However, it remains unclear to what extent INPs transferred from the ocean into the atmosphere contribute to the INP pool in the atmosphere, since further studies have identified other sources of INPs besides sea spray aerosol (Gong et al. (2020), and references therein).

## 4.   Conclusion

At the CVAO, concerted measurements of lipids as representatives of their respective classes were performed during the MarParCloud campaign to determine their concentrations in seawater and SML (as dissolved and particulate lipids) and on submicron aerosol particles. In seawater, the detailed analysis of the lipid classes showed that although the concentrations of PL and DL are generally very similar, the composition of the lipids to the PL and DL groups exhibits several differences. This suggests that different production and degradation mechanisms for DL and PL contribute to the respective lipid composition.

On the aerosol particles, the lipid composition resembles the lipid composition of the dissolved fraction in seawater. Although the lipids are reported as fast reactive compounds, our results suggest that they are more stable in the DL and PL in ULW and more degraded in the PL in SML. The phytoplankton groups chlorophytes, cyanobacteria and diatoms probably influence the lipid abundance as shown by pigment measurements. However, the concentration of chl-*a*, as often used proxy for biological production via phytoplankton, is not sufficient to describe the lipid concentration. Our results indicate that, besides



phytoplankton, bacteria also play an important role in lipid abundance in the oligotrophic North Atlantic, as shown by the PE/PG ratio and the abundance (and slight SML enrichment) of TCN. The concentration and enrichment of lipids in the ambient SML is not related to their physico-chemical properties describing the surface activity, at least not in the short term, probably due to parallel in-situ formation and degradation processes. This is underlined by the fact that lipids in the top 100

µm of the SML are not as highly enriched as other (less surface active) compound, such as amino acids.

For aerosol, however, the high enrichment of total lipids corresponds well with the consideration of their high surface activity. The $EF_{aer}$ agrees with the considerations from modelling and laboratory studies that, among the marine OM groups, lipids are the most highly enriched compounds and indicates that the transfer of lipids from the SML to the aerosol phase in the complex marine field is consistent with processes described in laboratory studies. In addition, the $EF_{aer}$ of lipids was one to two orders

of magnitude higher than the $EF_{aer}$ of the less surface-active amino acids previously reported. In terms of the individual lipid groups on the aerosol particles, a mild relation between $EF_{aer}$ and lipophilicity (expressed by the $K_{OW}$ value) was observed, which was missing in SML. In general, however, the parameters representing the surface activity of the lipid classes (density, $K_{OW}$ value and TPSA) were not sufficient to describe their transfer to the aerosol particles. The fact that bacteria are strongly involved in lipid abundance underlines that models using chl-*a* are not enough to describe OM in general and lipids in

particular. In addition, physico-chemical OM properties such as surface activity, are not sufficient to describe lipid abundance in the complex marine environment. Further processes such as biotic formation and degradation, as shown by the investigation of the individual lipid classes, contribute to lipid abundance in seawater, SML and on aerosol particles in the marine environment and must be included in the consideration of lipid transfer and finally in OM transfer models. Beyond that, our data suggest that the enrichment of the lipid classes on aerosol particles may be related to the distribution of the lipid on their

respective adsorption coefficients in water ($K_{aq}$) and in air ($K_a$). Compounds which are preferably arranged within the bubble interface ($K_{aq} \sim K_a$), namely TG and ALC, are transferred to the aerosol particles to the highest extend. Finally, our results showed that lipids had the potential to contribute to (biogenic) INP activity when transferred to the atmosphere.

Altogether, we showed that the diverse group of lipids represent an important and complex OM group in seawater and on marine aerosol particles. To the best of the author's knowledge, the present study is the first to analyze several lipid classes

simultaneously in seawater including ULW and SML and on submicron aerosol particles ($PM_1$) in such detail to obtain indications on their sources and sea-air linkage in the marine environment.

*Data availability.* The data are currently uploaded to the AWI World Data Centre PANGAEA (https:\\ww.pangaea.de/).

*Acknowledgements.*

This work was funded by Leibniz Association SAW in the project 'Marine biological production, organic aerosol particles and marine clouds: a Process Chain (MarParCloud)' (SAW-2016-TROPOS-2) and within the Research and Innovation Staff Exchange EU project MARSU (69089). S. Frka and B. Gašparović acknowledge the Croatian Science Foundation for the full support under the Croatian Science Foundation project IP-2018-01-3105 (BiREADI). The authors also thank Susanne Fuchs,





Anett Dietze, René Rabe and Anke Rödger for providing additional data and filter samples and all MarParCloud and MARSU project partners for a good cooperation and support. The authors thank Kerstin Lerche from the Helmholtz-Zentrum für Umweltforschung GmbH-UFZ in Magdeburg for the pigment measurements. Additional thanks to Joanna Waniek, Jenny Jeschek and Christian Burmeister for analysing inorganic nutrients and Katja Kaeding for flow cytometry measurements.

*Author contributions.* N. Triesch wrote the manuscript with contributions from M. van Pinxteren, S. Frka, B. Gašparović, C. Stolle, T. Spranger, E.H. Hoffmann, X. Gong, H. Wex, D. Schulz-Bull and H. Herrmann. N. Triesch and M. van Pinxteren performed the field sampling as part of the MarParCloud campaign team. N. Triesch supported by T. Spranger did the sample preparation for lipid measurements and N. Triesch, together with S. Frka and B. Gašparović, performed the lipid measurements. The lipid data evaluation was done by N. Triesch, S. Frka and B. Gašparović in consultation with M. van Pinxteren and H. Herrmann. C. Stolle and D. Schulz-Bull performed the (micro)biological and nutrient investigations in seawater and X. Gong and H. Wex the measurements of INP activity. The statistical analysis was done by N. Triesch in consultation with T. Spranger, C. Stolle, M. van Pinxteren and H. Herrmann. The introduction and implementation of the adsorption coefficients were carried out by N. Triesch with the support of E. H. Hoffmann and in consultation with M. van Pinxteren and H. Herrmann. All authors discussed the results and further analysis after the campaign. All co-authors proofread and commented the manuscript.

*Competing interest.* The authors declare that they have no conflict of interest.

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
