# Peer review of "Concerted measurements of lipids in seawater and on submicron aerosol particles at the Cape Verde Islands: biogenic sources, selective transfer and high enrichments"

_Atmospheric Chemistry and Physics, 2020_

## Referee Comment (RC1) · Anonymous Referee #1 · 15 Oct 2020

The manuscript presents a very interesting dataset on lipids, investigated in seawater, sea-surface microlayer and submicron aerosol particles at the CVAO – Cape Verde Atmospheric Observatory. Both dissolved and particulate lipids were studied, showing different degrees of enrichment and the partitioning into different classes among the three compartments studied. This is very interesting, and as the authors point out, the lipid composition in seawater, sea-surface microlayer and aerosols highlights that not only autotrophic sources may be responsible for the organic composition of submicron aerosols, but that bacterial activities and further oxidation and photodegradation pro-

cesses on these atmospheric particles may be responsible for their different organic matter content. This is important to individuate the sources of these materials and possibly, their behavior once in the atmosphere.

The manuscript is well written, just some minor spelling mistakes and wording should be looked at, and it well addresses the scope of ACP. It is also novel as it presents concerted measurements on three compartments, usually studied separately in field work. The scientific approach and methodology is sound and well described, accompanied with an extensive sub-set of supplementary information.

Overall I recommend the publication of the paper, but I suggest some minor issues to be taken into account prior to this.

I think the authors shall include some more information on the importance of this study in the field. I found the manuscript technically correct, but in my opinion it lacks a bit of background information and broader outlook. I have these comments in particular:

What is the importance of PM1 aerosols with respect to other size-distributions? How would you expect lipid classes be different in other size-fractions of marine aerosols?

Why did you chose the glass plate for SML sampling compared to other devices, and can you estimate (if any), biases of the glass-plate method on lipids concentration with respect to other components of this layer?

Line 32 page 5: It is not clear to me why did you analyze Na+ and why you used it to calculate EFaer. Is it related to seawater salts?

In the figures (e.g. figure 1), maybe you can specify in the caption that where there is no _SML you refer to ULW, I suppose.

Lines 7-11, page 11: it is not very clear to me whether in the SML PL were enriched, or DL, as in page 18, line 32, you state that PL are more degraded in the SML, and more stable in the DL and PL in the ULW. This sounds a bit confusing. If PL are enriched in the SML, I would expect them to be more stable then, and less degraded.

Line 23, page 11: is it possible that the lower enrichment of lipids compared to other OM classes like amino acids is due to biases in the glass plate method?

Line 3 page 12: it is also possible that phytoplankton activity is lower in the SML because of high radiation, thus cells stay preferentially below the SML. I expect that at the Cape Verde Islands solar radiation was quite high. Are there any measurements on this parameter? It would be interesting to see how it can influence the different lipid composition in seawater, SML and PM1. Do you have any data on this? Did you run some tests?

How do you relate the lipid composition of submicron aerosols, that have presumably been in the atmosphere for a longer residence time, to the ambient seawater composition both in ULW and SML, that may rapidly change even during diel cycles? I don't know if it makes sense, but would it be possible to estimate an EFaer based on ULW properties instead of SML components?

Just as a curiosity, would bigger aerosol particles show a more straightforward relation to seawater properties, considering that they may have resided in the atmosphere for a shorter time and travelled over shorter distances?

The PM1 lipid fraction resembled more the DL fraction of seawater, in composition. Would it be possible that by sampling larger aerosols, the lipid composition would have resembled more the PL fraction of seawater instead? If so, could this be because of molecular size?

What are the implications of lipid composition for marine aerosols? Could the different lipid fractions lead to different cloud forming capacities, or aerosol properties concerning optical thickness and radiative effects? Are these properties relevant for the study region, or other marine regions? I think it would be a good addition to the paper to discuss these aspects and broader implications in the conclusion section.

[Figure]

2020.

---

## Referee Comment (RC2) · Anonymous Referee #2 · 20 Oct 2020

General comments

The authors of the manuscript 'Concerted measurements of lipids in seawater and on submicron aerosol particles at the Cape Verde Islands: biogenic sources, selective transfer and high enrichments' present a valuable data set. The concerted measurement of a broad range of lipid classes in seawater, in the sea surface microlayer and on submicron aerosols is novel and benefits the scientific community as an inventory. This data bridges oceanic and atmospheric research by applying a common method and thus enabling a direct comparison between organic matter present in each realm.

This is a clear step forward into required interdisciplinarity within the field and fits the scope of ACP. Overall, I recommend to publish this paper.

However, several major improvements on representing and discussing this dataset can be made. In the introduction, a brief outline of the study area, i.e. the Tropical Atlantic, in terms of phytoplankton bloom dynamics, relevance for aerosol formation and ice nucleation activity should be given. It is not completely conclusive how the work ultimately relates to Chl-a as a proxy for in general phytoplankton biomass (?) or org. matter enrichment in aerosols, although discussed over some lines. The authors should formulate a clearer statement.

In general, it is crucial to discuss location of sampling and the temporal succession of sample events since enrichment factors for aerosols are calculated and, more importantly, conclusions on processes leading to the observed enrichment are drawn. Marine sources of the aerosols sampled at CVAO most likely do not match the location where they have sampled the SML and ULW, nor did sampling of aerosols (over the course of 24 hours) matches the specific time slots of seawater sampling.

The authors further compared the enrichments of lipid classes in the SML and aerosols to a defined theoretical 'surface activity' characterized by certain criteria i.e. density, partitioning coefficient between octanol and water (Kow) and topological polar surface area (TPSA). It should be kept in mind, that the solute is water and thus surface enrichment of lipids may be rather dictated by amphiphilic behavior i.e. an increase in TPSA and lower Kow. Also, it should be discussed how the two models of 'surface activity' compare (the authors also calculate adsorption coefficients based on concentrations and saturation vapor pressure as proxy for enrichment in the bubble-water-interface). I am not yet completely convinced that these approaches actually help to understand surface dynamics of enriched substances in the marine realm.

In the end, I advise that the authors should focus on the biological context since all lipid classes seem to relate to the marine realm, degradation indices are derived, pigment

analysis and basic abundances of microorganisms were measured and INP analysis ultimately shows that a strong biological component controls activity and aim to better link their findings.

Specific comments

Page 2 Line 24-26 Clarify: Marine dissolved lipids are produced either by dissolution from the particulate fraction, or 'by' primary production... living cells are also part of the particulate pool. Maybe better distinguish between abiotic and biotic processes and include the microbial loop?

Page3 Line 3 Consider quoting Becker et al. 2018 on TG's as storage compounds in phytoplankton.

Line 8-9 'However, Chl-a (concentration?) is also found to be a poor descriptor of autotrophs (biomass, cell abundance?), especially in oligotrophic regions (Quinn et al., 2014).' The authors should clarify this, since Quinn et al. concluded that Chl-a concentration is only a poor proxy for organic matter enrichment in aerosols.

Line 27-30 '...TG lipid class serves as an indication that the aerosol particles consist to a certain extent of freshly emitted sea spray...' Additional literature or an explanation would be very helpful, since Schiffer et al. 2018 concluded that on SSA surfaces the 'reduction in activity could essentially reduce the processing (by BC Lipase) of triacylglycerols into fatty aicds' i.e. if TG is present in SSA, it is not necessarily an indicator of freshly emittance. Also relevant on page 13, line 8.

Line 29 'In laboratory studies by the authors Schiffer et al. (2018), lipase enzymes have shown to be transferred from the ocean into the atmosphere...' Again, additional literature and explanations are needed. Schiffer et al. 2018 conducted a laboratory experiment on surface behavior of lipase and lipids in a Langmuir trough and conducted molecular dynamics simulations to judge on the activity of enzymes on SSA.

Page 4 Line 19 A map illustrating seawater sampling stations and CVAO location including distances and height of tower (!) would be helpful. Also, wind directions over the sampling period seem crucial to your study.

Page 5 Line 25-28 The authors should mention that the analysis was conducted by Flow Cytometry. It is not completely clear, if the analysis of eukaryotic (based on autofluorescence?) and prokaryotic cells (based on staining with SYBR green?) was conducted simultaneously and if autotrophic prokaryotes were excluded from prokaryotic cell numbers?

Page 6 Line 21 '. . . while lower LI values indicate that the appearing lipid classes are very fresh or resistant to degradation. . .' In my opinion, this is somehow critical and should be explained in more detail, since degradation products are themselves defined by their resistance to further degradation. This influences also concluding remarks later on, e.g. Page 10, line 2 '. . .suggesting that the dissolved lipid classes were quite resistant to degradation. . .' How can the authors decide whether lower LI's indicates fresh production or resistance to degradation as introduced in the experimental section?

Line 13 'However, these differences between bacterial and phytoplankton sources are not reflected in the total observed (particulate) lipid pool, because degradation products like FFA also contribute strongly.' Since FFA are present in the particulate fraction they apparently had to be enclosed within intact cells or other larger particles ($>0.7\mu$m). To my understanding, FFA would be part of the dissolved fraction otherwise. Thus, I am not so sure if FFA can serve as an indicator of degradation when encountered within the particulate pool. Is the LI defined as a proxy for degradation in the dissolved and particulate phase likewise (Goutx et al. 2003)?

Page 9 Line 15-16 The authors should quote, which lipid class they refer to when talking about 'chlorophyll degradation products'.

Line 18-19 Please clarify to what exactly you are referring to. Does 'This observation' relates to enhanced degradation in the SML or simply high LI values in the East Atlantic Ocean?

Page 10 Line 33 Since the authors judge on '. . .Chl-a as a proxy for bioproduction, may not sufficiently explain the variability of lipid classes. . .' They should introduce their results regarding Chl-a in greater detail instead of referring to a table in the supplementary material. Also, I do not recall an introduced scientific discussion concerning the reliability of Chl-a as a proxy for lipid classes.

Page 11 Line 7-14 '. . . slightly higher enrichment of the particulate fraction. . .' I actually do not think, this is meaningful to discuss in relation to the presented results, since variance of the dissolved EF's range within the larger variance of particulate EF's and means only very slightly.

Line 12-14 'Moreover, marine dissolved lipids can be produced by dissolution from the particulate fraction and through primary production and released during the life cycle and after cell death. This(!) might lead to a slightly higher SML enrichment of the particulate lipids.' Please elucidate, I cannot follow the conclusion made. Why does a dependence of the dissolved pool from the particulate pool indicate higher enrichments? Increased degradation and abiotic photochemical reaction within the SML could likewise produce higher enrichment of the dissolved fraction. . .

Page 15 32 Lead the reader towards your conclusion stating that 'a differentiation of the contribution' of the particulate versus the dissolved pool was not possible also when taking into account the size of the fractions. To my understanding it is more likely that the fraction of lipids smaller than $0.7\mu$m (i.e. dissolved) contributed to submicron aerosols (PM1).

Page 15 Line 14 I actually sense it is assumed that marine bacteria transmitted into the atmosphere behave similarly in terms of production and metabolism than within the hydrosphere i.e. their natural habitat. I think, this is a hypothesis which needs to be discusses more carefully. (Also Page 4, line 4)

Page 18 Line 19 '. . .samples are consistent with the results of Wilson et al. (2015) indicating that lipids in the tropical North Atlantic Ocean have. . .'. This could leave

the reader under the impression that Wilson et al. 2015 have assessed lipids and concluded they contribute to the biogenic INP pool.

Line 34 'However, concentration of Chl-a, as often used proxy for biological production via phytoplankton, is not sufficient to describe lipid concentration.' Again, Chl-a is not described as a proxy to determine lipid classes in literature.

Supplementary Material Page 28 Table S7 'XLogP3-AA' replaces Kow, which is found in the main text, yet for the method in use to calculate this value, no literature is provided.

Technical comments

Page 1 Title: 'Concerted measurements of lipids in seawater and on submicron aerosol particles at the Cape Verde Islands: biogenic sources, selective transfer and high enrichments'. The authors should overthink the title, e.g. include instead of 'high enrichment', 'ice nucleating potential' to better describe the content of the article.

Line 16-23 Exclude 'To this end'. The set of lipid classes analyzed includes ... and rephrase the following sentence: Introduced lipid classes have been analyzed in the dissolved and particulate fraction of seawater, while differentiating between underlying water (ULW) and the sea surface microlayer (SML), and on submicron aerosol particles (PM1) collected from the ambient (air?) at the Cape Verde Atmospheric Observatory (CVAO). Or consider other fragmentation.

Line 24 Include '$\sum$' $to\,align\,style\,to\,the\,rest\,of\,the\,text.$

Line 32-33: For aerosols, however, the high enrichment of lipids (as a sum) on aerosols corresponds well... Include 's' and exclude one of the redundant 'aerososls'.

Line 32 Separate 'physico-chemical' to align style to the rest of the text.

Page 2 Keywords: consider to replace rather generic words such as 'seawater', 'concerted measurements', 'transfer' by e.g. 'sea surface microlayer', 'sea spray aerosols'

to characterize the work.

Page 4 Line 6 Rephrase and clarify this sentence 'is discussed in terms of biological and physical (INP) parameters...' E.g. is discussed in the context of its biological origin and its ice nucleation potential.

Page 5 Line 33 The authors should briefly explain the unit in use: Does the unit relates to the total filter area used for the extraction of lipids in aerosols (28.27cm2)?

Page 7 Line 14 Exclude, since this is a repetition of line 12: '...but considered a 'trend' to be valid...'.

Page 8 Line 8 Maybe introduce the PE/PG ratio along with LI and EF's in the experimental section.

Line 10 Replace 'afterwards' with 'towards the end'.

Line 11 Consider rephrasing or exclude '-': 'This indicates a change in the lipid dominant biological contributions, with bacterial sources dominating in the first part and especially in the middle of the campaign, whereas in the last part rather phytoplankton-dominated contributions to the lipid pool.'

Page 9 Line 18 Include the articles '...release in the SML compared to the ULW...'.

Page 10 Line 32 Exclude 'the': '...it is most likely that the bacteria have influenced...' Check for consequential mistakes.

Page 11 Line 23 '...OM compound groups...' I think, this is redundant, use groups or compounds instead.

Line 28 Instead of 'regarded' use 'considered'.

Page 12 Line 6 '...different diversity and different taxa...' Redundant if diversity actually indicates the composition of species. However, it can be defined as functional etc.

Line 8 '. . .in the particulate fraction. In the particulate fraction. . .' Try to rephrase due to repetition.

Line 11 Consider rephrasing: This indicates that the lipid reserves are stored in the particulate lipids and are dissolved producing dissolved TG. For example: This indicates that lipid reserves such as TG are stored within the particulate pool and upon dissolution become part of the dissolved pool.

Line 13 Use 'physicochemical descriptors' instead of 'physical processes'.

Line 28 Caption of Fig. S11 states 'dissolved' lipids in aerosols particles, which is probably a mistake.

Page 13 Fig. 3 Absolute concentration of lipids in aerosol particles do not fit percentage data in the supplement of Fig.S11. For example, on the 29/09/2017 PE are present in Fig.3 while being completely absent in Fig. S11, the color schemes might have been confused.

Line 6 'bacteria, possibly transported from the ocean into the atmosphere, produce PE on aerosol particles'. . . Better to replace 'produce' by 'contribute'.

Line 10 Replace 'maritime samples' by 'of marine origin'.

Page 15 Line 9 This is misleading, better state 'aerosol particles' instead of 'particle phase'.

Line 30 Rephrase: 'The finding here, that both DL and PL, contain similar classes of lipid, which are also found on the aerosol particles, suggest that both types of lipids in seawater are transferred to the aerosol particles via bubble bursting process.'

Page 17 Figure 6 'interface' is hard to read. Improve color scheme. There is also a logical mistake, since the caption states 'Scheme of a bubble during the bubble bursting process'. During bubble bursting, the bubble actually has reached the air-water interface i.e. exhibits two surfaces oriented towards the air inside and the atmosphere

outside. Otherwise, the caption should state 'during the process of a bubble rising through the water column'...

Line 12 Rephrase 'contain... abilities'.

Page 18 Line 11-12 Replace 'by...' with 'of sea spray aerosols'.

Line 25 Consider rephrasing: 'At the CVAO, concerted measurements of lipids as representatives of their respective classes were performed during the MarParCloud campaign to determine their concentrations in seawater and SML (as dissolved and particulate lipids) and on submicron aerosol particles.' For example: Concerted measurements of lipids were performed in proximity to the Cape Verde Islands to compare the concentration of specific lipid classes in submicron aerosol particles and in the dissolved and particulate phase of seawater (ULW and SML).

Line 27 Consider rephrasing: E.g. The analysis of lipid classes in seawater showed that, although concentrations in the particulate and dissolved phase are generally very similar, the contribution of lipids within phases differed.

Page 23 Line 31-35 Check format, looks like a line spacing error.

Page 25 Line 1 Adjust the predicate 'Van' to the same format i.e. 'van'.

Supplementary Material I recommend to shorten the supplementary information provided, maybe consider excluding Fig. S7, 8, 11, S12, S13, S19.

Equalize color scheme and figures i.e. when drawing a regression line, use the same design and report same correlation values as in the main text e.g. R versus R2

---

## Author Comment (AC1) · 10 Dec 2020

We thank the reviewer for the careful examination of the manuscript and the supporting information. In the document 'author's response - acp-2020-432-Referee 1', please find a point-by-point response to the questions and concerns. All references to the manuscript (e.g. page and line numbers) listed in our replies refer to the clean version of the now revised manuscript (without track changes).

Please also note the supplement to this comment:

[Figure]

https://acp.copernicus.org/preprints/acp-2020-432/acp-2020-432-AC1-supplement.pdf

[Figure]

**Supplement:**

The manuscript presents a very interesting dataset on lipids, investigated in seawater, sea-surface microlayer and submicron aerosol particles at the CVAO – Cape Verde Atmospheric Observatory. Both dissolved and particulate lipids were studied, showing different degrees of enrichment and the partitioning into different classes among the three compartments studied. This is very interesting, and as the authors point out, the lipid composition in seawater, sea-surface microlayer and aerosols highlights that not only autotrophic sources may be responsible for the organic composition of submicron aerosols, but that bacterial activities and further oxidation and photodegradation pro-cesses on these atmospheric particles may be responsible for their different organic matter content. This is important to individuate the sources of these materials and possibly, their behavior once in the atmosphere.

The manuscript is well written, just some minor spelling mistakes and wording should be looked at, and it well addresses the scope of ACP. It is also novel as it presents concerted measurements on three compartments, usually studied separately in field work.

The scientific approach and methodology is sound and well described, accompanied with an extensive sub-set of supplementary information.

Overall I recommend the publication of the paper, but I suggest some minor issues to be taken into account prior to this. I think the authors shall include some more information on the importance of this study in the field. I found the manuscript technically correct, but in my opinion it lacks a bit of background information and broader outlook.

We thank the reviewer for the careful examination of the manuscript and the supporting information. In the following, please find a point-by-point response to the questions and concerns. All references to the manuscript (e.g. page and line numbers) listed in our replies refer to the clean version of the now revised manuscript (without track changes).

I have these comments in particular:

R#1-1 a) What is the importance of PM1 aerosols with respect to other size-distributions?

Submicron aerosol particles ($PM_1$) are in a size range most important for cloud processes, e.g. play a critical role in the formation of cloud condensation nuclei (Quinn and Bates, 2011) and have an atmospheric lifetime of several days (Madry et al., 2011). Moreover, marine aerosol particles contain a large quantity of organic material (e.g. Quinn and Bates (2011) and references therein). The enrichment of organic matter increases with particle size (O'Dowd et al., 2004), hence, the submicrometer aerosol particles are often dominated by organic matter (OM) (Quinn and Bates, 2011).

R#1-1b) How would you expect lipid classes be different in other size-fractions of marine aerosols?

Investigations of lipids on marine aerosol particles, especially in different size ranges, are still sparse.

In a laboratory mesocosm experiment, Cochran et al. (2016) investigated the fatty acid composition in sub- and supermicron aerosol particles and reported that about 75% of the submicron aerosol particles showed strong signals for the presence of long-chain fatty acids, whereas supermicron sea spray aerosol particles were dominated (up to 88%) by oxygen-rich species. The study of Cochran et al. (2016) shows the large varying composition of lipids in (size-resolved) aerosol particles generated in the lab. Here we aimed to contribute the composition and concentrations of lipids in ambient marine aerosol particles and discuss their relations and we choose $PM_1$ aerosol particles as explained above.

These aspects regarding the importance of $PM_1$ aerosol particles have briefly been discussed in the Introduction of the manuscript.

In the revised version we added on page 3, line 31 - page 4, line 2: "Specific lipid classes such as long chain fatty acids and cholesterol as constituents of aerosol particles are already regarded as important factors for the activation of aerosol particles to cloud condensation nuclei (CCN) (Barati et al., 2019) or ice nucleation particles (INP) (Nguyen et al., 2017;DeMott et al., 2018). Cochran et al. (2016b) investigated the fatty acid composition in sub- and supermicron sea spray aerosol particles and reported that about 75 % of the submicron aerosol particles showed strong signals for the presence of long-chain fatty acids. In contrast, supermicrometer sea spray aerosol particles were dominated (up to 88 %) by oxygen-rich species (Cochran et al., 2016b)."

R#1-2) Why did you chose the glass plate for SML sampling compared to other devices, and can you estimate (if any), biases of the glass-plate method on lipids concentration with respect to other components of this layer?

The glass-plate technique in general:
A strong advantage of using the glass plate technique is that a thin SML sample (usually 20-150 μm thickness) is collected and the biological composition of the SML is more representative (Cunliffe, 2014). Another advantage is the simple way of sampling and the easy-to-use-format (important in field studies). A disadvantage of this sampling method is the time consuming sampling (~45 min to collect 1 L sample)(Cunliffe, 2014). Altogether, the glass plate method for collecting SML is an established sampling technique that has often been used in previous studies (e.g. Reinthaler et al. (2008);Wurl and Holmes (2008);Engel and Galgani (2016);Zäncker et al. (2018);van Pinxteren et al. (2017)).

The glass-plate technique regarding lipids:
During this study, we exclusively used the glass-plate technique and cannot compare the sampling efficiency of this technique towards others for lipids. Early papers have speculated that the glass-plate might not be very effective for hydrophobic lipids, as glass itself has a hydrophobic surface (e.g. van Vleet and Williams (1980)). However, more recent work shows that the glass-plate technique is very well suited to collect highly hydrophobic dissolved organic substances such as lipids and amino acids, e.g. Cunliffe (2014), Stolle et al. (2019). Therefore, we have no reason to believe that the glass-plate technique is more prone to biases than others. We carefully tests contamination and carry over problems by taking blanks as described in section 2.2.1.

In the revised version of the manuscript we have added the following (page 5, line 2-4): "The SML samples (n=6) were collected using the manual glass-plate technique, a standard SML sampling method whose correct application and specification are described in detail in the 'Guide to best practices to study the ocean's surface' by Cunliffe (2014)."

R#1-3) Line 32 page 5: It is not clear to me why did you analyze Na+ and why you used it to calculate EFaer. Is it related to seawater salts?

The Na$^+$ concentration in both, seawater and aerosol particles, are necessary to calculate the EF$_{aer}$ that is a quantitative metric for the comparison of compounds in the ocean and in the atmosphere. It considers the analyte concentration in the different matrices (seawater, aerosol particles) in relation to the Na$^+$ concentration in both matrices.

In the revised version we added a more detailed explanation of the EF$_{aer}$ in section 2.2.4 'Enrichment factors', which reads as follows (page 7, line 22-26): "The EF$_{aer}$ is a quantitative metric for the comparison of compounds in the ocean and in the atmosphere. The EF$_{aer}$ concept is mainly applied to closed systems (Quinn et al. (2015) and references therein, Rastelli et al. (2017)) since formation or degradation pathways on aerosol particles including biological or photochemical atmospheric reactions and possible transports from other than marine sources are excluded for this parameter. However, for comparison purposes it is useful to calculate the EF$_{aer}$ also for open systems, as in the studies of e.g. Russell et al. (2010) or van Pinxteren et al. (2017)."

R#1-4) In the figures (e.g. figure 1), maybe you can specify in the caption that where there is no _SML you refer to ULW, I suppose.

We have now more clearly distinguished between SML and ULW samples in the figures of the seawater samples in the revised version. The SML samples are specially highlighted in the figures, e.g. 20/09/2017_SML, while the ULW samples are only described as sampling date (e.g. 20/09/2017). For easier differentiation between ULW and SML samples, we have improved the labelling of the figures (Fig. 1 and Fig. 2).

The caption of Fig. 1 reads now as follows (Manuscript, page 8): "Figure 1: Concentration of individual particulate lipid classes in the ULW (sampling date) and the SML (sampling date_SML) samples along the campaign and as an averaged value in µg L$^{-1}$."

R#1-5) Lines 7-11, page 11: it is not very clear to me whether in the SML PL were enriched, or DL, as in page 18, line 32, you state that PL are more degraded in the SML, and more stable in the DL and PL in the ULW. This sounds a bit confusing. If PL are enriched in the SML, I would expect them to be more stable then, and less degraded.

These statements refer to the results achieved from a) the EF of PL and DL groups in the SML and b) the LI. The EF describes the enrichment/depletion of the entire DL or PL group in the SML and the LI is the ratio between the metabolites (ALC, FFA, MG, DG) and intact lipids (TG, WE, glycolipids as MGDG, DGDG, SQDG and phospholipids as PG, PE, PC) as described in equation 1 (Manuscript, page 6). As these two parameters regard different compound classes/ratios, it is no contradiction that in the SML compounds are enriched, but at the same time degradation can be interpreted from the ratio of metabolites and intact lipids, the LI.

Taking this comment into account, we have carefully rewritten the interpretations of the Lipolysis Index as follows in the revised manuscript:
In section '2.2.3 Lipid ratios' it now reads as follows: "Higher LI values are characteristic for enhanced OM degradation and metabolite release, while lower LI values indicate that the appearing lipid classes are more fresh or resistant to degradation." (page 7, line 5-7)
In section 3.1.3 it now reads: "However, on specific days, the LI$_{SML}$ of PL was ≥ 0.5 (Table S5), indicating a slightly increased OM/lipid degradation and metabolite release in the SML compared to the ULW." (page 10, line 17/18)
and "The LI of DL (Table S5) varied between 0.13-0.53 in the ULW and between 0.20-0.48 in the SML samples, suggesting that the dissolved lipid classes were somewhat more resistant to degradation." (page 10, line 20 - page 11, line 2)

and in the Conclusion: "Although the lipids are reported as fast reactive compounds, our results suggest that the DL are somewhat more resistant to degradation." (page 19, line 30-31)

R#1-6) Line 23, page 11: is it possible that the lower enrichment of lipids compared to other OM classes like amino acids is due to biases in the glass plate method?

The glass-plate technique is not known to be more prone to biases compared to other SML sampling techniques. We also measures amino acid samples with the same technique at the same location and found higher enrichment factors for them (Triesch et al., 2020). In this context, we would like to refer to the reviewer's comment R#1-2 regarding the SML sampling technique using the glass-plate method and its advantages and disadvantages.

R#1-7) Line 3 page 12: it is also possible that phytoplankton activity is lower in the SML because of high radiation, thus cells stay preferentially below the SML. I expect that at the Cape Verde Islands solar radiation was quite high. Are there any measurements on this parameter? It would be interesting to see how it can influence the different lipid composition in seawater, SML and PM1. Do you have any data on this? Did you run some tests?

The solar radiation during the campaign was measured with a 'Pyranometer SKS 1110' (Skye Instruments Ltd, Powys, United Kingdom) installed on the 10 m high tower of the Cape Verde Atmospheric Observatory (CVAO).
The table shown here shows the averaged solar radiation data (243.3 – 676.2 W m$^{-2}$) over the sampling period of the SML samples. Although a variance of the solar radiation data could be observed, no statistically relevant correlation/trend between the solar radiation data and the SML lipid concentrations or composition could be found, which was probably due to the small number (#5) of corresponding sample numbers.

| Sampling time local time | Average solar radiation [W m$^{-2}$] during the sampling time |
|---|---|
| 25/09/2017 9:45-10:48 | 676.2 |
| 27/09/2017 8:50-10:03 | 581.7 |
| 06/10/2017 8:04-9:47 | 371.2 |
| 07/10/2017 09:22-10:35 | 551.4 |
| 10/10/2017 8:30-9:30 | 243.3 |

The SML samples were always collected in the morning between 8:04 and 10:35 local time (UTC-1) on different days (for more details, see van Pinxteren et al. (2020)). Due to the limited sampling possibilities in Cape Verde (SML sampling from a small fishing boat), it was not possible to carry out seawater sampling at other times of the day, e.g. in the evening/night period or even investigate diurnal cycles. The aerosol particle samples are 24 h samples, hence both day and nighttime influences on aerosol lipid composition (e.g. solar radiation), the sampling periods would have to be adjusted. We will take this interesting aspect of solar radiation into future campaign.

In the revised supporting information, we included we solar radiation measurements in Table S8.

R#1-8 a) How do you relate the lipid composition of submicron aerosols, that have presumably been in the atmosphere for a longer residence time, to the ambient seawater composition both in ULW and SML, that may rapidly change even during diel cycles?

In our approach of concerted measurements, we compared PM$_1$ aerosol particles, sampled for 24 h with spot samples taken in the ocean (ULW, SML) within the aerosol sampling period. To allow a comparison of these two matrices, we strongly considered several additional measurements, such as backward-trajectories, the concentrations of inorganic ions and mineral dust tracers on the aerosol particles measured during the campaign. These parameters were discussed in detail in the overview paper of the campaign (van Pinxteren et al., 2020) and in a separate paper measuring amino acids within this campaign and influences (Triesch et al., 2020).

We have added these considerations in a new subchapter '3.2.1 The comparability of the different marine matrices (seawater and aerosol particles)' of section '3.2 Transfer of lipids from the Oceans'. It reads now as follows (page 12, line 6-11): "The concerted measurements performed here included spot samplings in the ocean (ULW, SML) during the sampling period of PM1 aerosol particles at the CVAO (24h). The air masses arriving at the CVAO often followed the water current (Peña-Izquierdo et al., 2012;van Pinxteren et al., 2017) and suggest an enhanced link between the upper ocean and the aerosol particles, as mainly winds drive the ocean currents in the upper 100 m of the ocean. The backward trajectories as well as the concentrations of inorganic ions and mineral dust tracers on the aerosol particles measured during the campaign, suggested a predominantly marine origin with low to medium dust influences (Triesch et al., 2020;van Pinxteren et al., 2020)."

We could not investigate diel cycles and would like to refer to the reviewer comment R#1-7.

R#1-8b) I don't know if it makes sense, but would it be possible to estimate an EFaer based on ULW properties instead of SML components?

The referee rightly stated that the EF$_{aer}$ can also be calculated based on the ULW concentrations. This has already been done in a previous study on amino acids by Triesch et al. (2020). Since the lipid concentrations in ULW and SML were quite similar and this is reflected in only comparatively small enrichments (EF$_{SML}$: 1.0-1.7), the calculated EF$_{aer}$ (based on ULW and based on SML) are also very similar. The EF$_{aer}$ based on SML concentrations is on average $2.6 \cdot 10^5$ and the EF$_{aer}$ based on ULW is on average $3.4 \cdot 10^5$ and therefore agree well. Thus, the calculation of the EF$_{aer}$ based on ULW does not provide any new insights compared to the EF$_{aer}$ based on SML and therefore we would prefer not to elaborate on this in the manuscript.

R#1-9) Just as a curiosity, would bigger aerosol particles show a more straightforward relation to seawater properties, considering that they may have resided in the atmosphere for a shorter time and travelled over shorter distances?

This is an interesting thought; however, probably complex (selective) transfer processes of lipids travelling from the ocean to the atmosphere and subsequent reactions determine the lipid composition of ambient aerosol particles in the sub- and supermicron range.
In a recent study from the same location, we could show that amino acids showed a higher diversity in submicrometer aerosol particles compared to supermicron particles. However, the composition of the amino acids in the submicron aerosol particles was more similar to that of seawater than to the supermicron aerosol particles (Triesch et al., 2020).

In the manuscript, we reported similarities between the composition of lipids in seawater and on the aerosol particles (e.g. section '3.2.3 Transfer of lipid classes from the ocean to the aerosol particles'). However, as we solely performed measurements of submicron aerosol particles it remains speculative, whether the lipid composition of the supermicron aerosol particles is more similar to that of seawater. This is certainly an interesting though to be addressed in future campaigns.

R#1-10) The PM1 lipid fraction resembled more the DL fraction of seawater, in composition. Would it be possible that by sampling larger aerosols, the lipid composition would have resembled more the PL fraction of seawater instead? If so, could this be because of molecular size?

Since we have only examined $PM_1$ aerosol particles, a statement about the lipid composition of larger aerosol particles is highly speculative. We would like to point out the sample preparation for lipid analysis in seawater and on the aerosol particles: The seawater was divided into the dissolved fraction (<0.7 µm) and the particulate fraction (0.7-200 µm). The aerosol particles (<1 µm) were collected as $PM_1$ aerosol particles. Furthermore, it must be considered that in the bubble bursting process, the formed droplets are water drops, which gradually dries up and finally leads to the formation of the aerosol particles. Therefore, the size separation in seawater is not transferable 1:1 to the aerosol particles.

Please see also our replies to R#1-1 and R#1-9.

R#1-11) What are the implications of lipid composition for marine aerosols? Could the different lipid fractions lead to different cloud forming capacities, or aerosol properties concerning optical thickness and radiative effects? Are these properties relevant for the study region, or other marine regions? I think it would be a good addition to the paper to discuss these aspects and broader implications in the conclusion section.

Lipid classes can serve as specific markers for the identification of OM sources, also with regard to possible biogenic sources and relations to (micro)organisms (see manuscript: page 2, line 26 - page 3, line 3 and page 19, line 31-34). Moreover, an important aspect of lipid composition for marine aerosol particles is the fact that lipid classes such as long-chain fatty acids can act as important factors for the activation of aerosol particles to CCN or INP as described in the manuscript (page 3, line 31-33). We would like to point out that the correlations found between lipids and ice nucleation potential relate to investigations in seawater as discussed in detail in reviewer comment R#2-25. It is difficult to extrapolate a transfer into aerosol particles with respect to organic compounds and ice nucleation potential, since other possible aerosol particles sources must also be considered when considering aerosol particles with respect to INP activity.

Statements regarding lipid fractions and various cloud forming capacities or aerosol properties such as optical thickness and radiative effects would be purely speculative, since no measurements on these properties of aerosol particles have been made that could be referred to. Due to these facts, we would like to limit our conclusions to the results that we were able to show and interpret by measurements in this study.

**Additional changes performed by the authors**

The acknowledgement was also revised to thank the people from the OSCM. The added sentence is now as follows: "We further acknowledge the professional support provided by the Ocean Science Centre Mindelo (OSCM) and the Instituto do Mar (IMar)." (page 21, line 4-6)

The measured data were published on PANGAEA. The data availability statement was therefore updated and reads as follows: "Data availability. The data are available through the World Data Centre

PANGAEA under the following link: https://doi.pangaea.de/10.1594/PANGAEA.921832." (page 20, line 27/28)

The previous citation of van Pinxteren et al. (2019) was updated to van Pinxteren et al. (2020) in the revised manuscript and supporting information

**References**

Cochran, R. E., Laskina, O., Jayarathne, T., Laskin, A., Laskin, J., Lin, P., Sultana, C., Lee, C., Moore, K. A., Cappa, C. D., Bertram, T. H., Prather, K. A., Grassian, V. H., and Stone, E. A.: Analysis of Organic Anionic Surfactants in Fine and Coarse Fractions of Freshly Emitted Sea Spray Aerosol, Environ. Sci. Technol., 50, 2477-2486, 10.1021/acs.est.5b04053, 2016.

Cunliffe, M. a. W., O.: Guide to best practices to study the ocean's surface. , Plymouth, UK, Marine Biological Association of the United Kingdom for SCOR, 118pp., 2014.

Engel, A., and Galgani, L.: The organic sea-surface microlayer in the upwelling region off the coast of Peru and potential implications for air-sea exchange processes, Biogeosciences, 13, 989-1007, 10.5194/bg-13-989-2016, 2016.

Madry, W. L., Toon, O. B., and O'Dowd, C. D.: Modeled optical thickness of sea-salt aerosol, Journal of Geophysical Research: Atmospheres, 116, https://doi.org/10.1029/2010JD014691, 2011.

O'Dowd, C. D., Facchini, M. C., Cavalli, F., Ceburnis, D., Mircea, M., Decesari, S., Fuzzi, S., Yoon, Y. J., and Putaud, J. P.: Biogenically driven organic contribution to marine aerosol, Nature, 431, 676-680, 10.1038/nature02959, 2004.

Quinn, P. K., and Bates, T. S.: The case against climate regulation via oceanic phytoplankton sulphur emissions, Nature, 480, 51, 10.1038/nature10580, 2011.

Reinthaler, T., Sintes, E., and Herndl, G. J.: Dissolved organic matter and bacterial production and respiration in the sea-surface microlayer of the open Atlantic and the western Mediterranean Sea, Limnol. Oceanogr., 53, 122-136, 10.4319/lo.2008.53.1.0122, 2008.

Stolle, C., Ribas-Ribas, M., Badewien, T. H., Barnes, J., Carpenter, L. J., Chance, R., Damgaard, L. R., Quesada, A. M. D., Engel, A., Frka, S., Galgani, L., Gašparović, B., Gerriets, M., Mustaffa, N. I. H., Herrmann, H., Kallajoki, L., Pereira, R., Radach, F., Revsbech, N. P., Rickard, P., Saint, A., Salter, M., Striebel, M., Triesch, N., Uher, G., Upstill-Goddard, R. C., Pinxteren, M. v., Zäncker, B., Zieger, P., and Wurl, O.: The MILAN campaign: Studying diel light effects on the air-sea interface, Bull. Amer. Meteorol. Soc., null, 10.1175/bams-d-17-0329.1, 2019.

Triesch, N., van Pinxteren, M., Engel, A., and Herrmann, H.: Concerted measurements of free amino acids at the Cape Verde Islands: High enrichments in submicron sea spray aerosol particles and cloud droplets, Atmos. Chem. Phys. Discuss., 2020, 1-24, 10.5194/acp-2019-976, 2020.

van Pinxteren, M., Barthel, S., Fomba, K. W., Muller, K., von Tumpling, W., and Herrmann, H.: The influence of environmental drivers on the enrichment of organic carbon in the sea surface microlayer and in submicron aerosol particles - measurements from the Atlantic Ocean, Elementa-Sci. Anthrop., 5, 21, 10.1525/elementa.225, 2017.

van Pinxteren, M., Fomba, K. W., Triesch, N., Stolle, C., Wurl, O., Bahlmann, E., Gong, X., Voigtländer, J., Wex, H., Robinson, T. B., Barthel, S., Zeppenfeld, S., Hoffmann, E. H., Roveretto, M., Li, C., Grosselin, B., Daële, V., Senf, F., van Pinxteren, D., Manzi, M., Zabalegui, N., Frka, S., Gašparović, B., Pereira, R., Li, T., Wen, L., Li, J., Zhu, C., Chen, H., Chen, J., Fiedler, B., von Tümpling, W., Read, K. A., Punjabi, S., C. Lewis, A. C., Hopkins, J. R., Carpenter, L. J., Peeken, I., Rixen, T., Schulz-Bull, D., Monge, M. E., Mellouki, A., George, C., Stratmann, F., and Herrmann, H.: Marine organic matter in the remote environment of

the Cape Verde Islands - An introduction and overview to the MarParCloud campaign, Atmos. Chem. Phys. Discuss., 2019, 1-63, 10.5194/acp-2019-997, 2019.

van Pinxteren, M., Fomba, K. W., Triesch, N., Stolle, C., Wurl, O., Bahlmann, E., Gong, X., Voigtländer, J., Wex, H., Robinson, T. B., Barthel, S., Zeppenfeld, S., Hoffmann, E. H., Roveretto, M., Li, C., Grosselin, B., Daële, V., Senf, F., van Pinxteren, D., Manzi, M., Zabalegui, N., Frka, S., Gašparović, B., Pereira, R., Li, T., Wen, L., Li, J., Zhu, C., Chen, H., Chen, J., Fiedler, B., von Tümpling, W., Read, K. A., Punjabi, S., Lewis, A. C., Hopkins, J. R., Carpenter, L. J., Peeken, I., Rixen, T., Schulz-Bull, D., Monge, M. E., Mellouki, A., George, C., Stratmann, F., and Herrmann, H.: Marine organic matter in the remote environment of the Cape Verde islands – an introduction and overview to the MarParCloud campaign, Atmos. Chem. Phys., 20, 6921-6951, 10.5194/acp-20-6921-2020, 2020.

van Vleet, E. S., and Williams, P. M.: Sampling sea surface films: A laboratory evaluation of techniques and collecting materials, Limnol. Oceanogr., 25, 764-770, 10.4319/lo.1980.25.4.0764, 1980.

Wurl, O., and Holmes, M.: The gelatinous nature of the sea-surface microlayer, Mar. Chem., 110, 89-97, 10.1016/j.marchem.2008.02.009, 2008.

Zäncker, B., Cunliffe, M., and Engel, A.: Bacterial Community Composition in the Sea Surface Microlayer Off the Peruvian Coast, Front Microbiol, 9, 2699-2699, 10.3389/fmicb.2018.02699, 2018.

---

## Author Comment (AC2) · 10 Dec 2020

We thank the reviewer for the careful examination of the manuscript and the supporting information. In the document 'author's response - acp-2020-432-Referee 2', please find a point-by-point response to the questions and concerns. All references to the manuscript (e.g. page and line numbers) listed in our replies refer to the clean version of the now revised manuscript (without track changes).

Please also note the supplement to this comment:

[Figure]

https://acp.copernicus.org/preprints/acp-2020-432/acp-2020-432-AC2-supplement.pdf

**[ACPD](ACPD)**

Interactive
comment

[Figure]

**Supplement:**

General comments

The authors of the manuscript 'Concerted measurements of lipids in seawater and on submicron aerosol particles at the Cape Verde Islands: biogenic sources, selective transfer and high enrichments' present a valuable data set. The concerted measurement of a broad range of lipid classes in seawater, in the sea surface microlayer and on submicron aerosols is novel and benefits the scientific community as an inventory. This data bridges oceanic and atmospheric research by applying a common method and thus enabling a direct comparison between organic matter present in each realm. This is a clear step forward into required interdisciplinarity within the field and fits the scope of ACP. Overall, I recommend to publish this paper.

We thank the reviewer for the careful examination of the manuscript and the supporting information. In the following, please find a point-by-point response to the questions and concerns. All references to the manuscript (e.g. page and line numbers) listed in our replies refer to the clean version of the revised manuscript (without track changes).

R#2-1) However, several major improvements on representing and discussing this dataset can be made.
R#2-1 a) In the introduction, a brief outline of the study area, i.e. the Tropical Atlantic, in terms of phytoplankton bloom dynamics, relevance for aerosol formation and ice nucleation activity should be given.

We thank the reviewer for the suggestions for improvement. In order to describe and to better motivate the study area, the tropical Atlantic, we have carefully revised section 2.1 'Study area and sampling sites'.
In the revised version we have stronger specified the marine region according to Longhurst (2007). The region investigated here, the 'North Atlantic Tropical Gyral Province (NATR)' region around the Cape Verde Islands, is an interesting but rarely studied oligotrophic region. The region often experiences clean marine air and low anthropogenic influences. Regarding the phytoplankton bloom dynamics, the NATR regions is described as an oligotrophic region with the lowest surface concentrations of chlorophyll in the North Atlantic and greater annual variability than seasonality (Longhurst, 2007). Regarding the topic of aerosol formation and ice nucleation activity, we included the finding that high marine INP concentrations were predicted in oceanic regions surrounding the Cape Verde Islands, in fact they were higher than in the rest of the North Atlantic (Wilson et al., 2015).

We added the information on the relevance of this region in terms of aerosol formation and ice nucleation activity in more detail in section 2.1 and it reads now as follows (page 4, line 24-29): "The ocean around the Cape Verde Islands belongs to the region 'North Atlantic Tropical Gyral Province (NATR)' according to the classification of Longhurst (2007), a region with the lowest surface chlorophyll in the North Atlantic and with a greater annual variability than seasonality. Wilson et al. (2015)

reported that high concentrations of marine INPs can occur in the North Atlantic and comparatively high surface-level marine (INP$_{15}$) and OC concentrations have been predicted by models in this region around the Cape Verde Islands."

R#2-1b) It is not completely conclusive how the work ultimately relates to Chl-a as a proxy for in general phytoplankton biomass (?) or org. matter enrichment in aerosols, although discussed over some lines. The authors should formulate a clearer statement.

In most parameterizations the transfer of OM from the ocean into the atmosphere and the prediction of the OM content on marine aerosol particles, is based on chl-$a$ seawater concentrations that are used as a broad indicator of biological productivity (Gantt et al., 2011;Rinaldi et al., 2013). However, chl-$a$ concentration alone does not adequately describe the complete spectrum of biological activity (Quinn et al., 2014) and especially in oligotrophic regions other parameters besides wind speed and chl-$a$ must be taken into consideration for a good prediction of OM on marine aerosol particles (van Pinxteren et al., 2017). Moreover, different groups of OM, such as lipids, carbohydrates and proteins show different characteristics in terms of their sea-air transfer (Burrows et al., 2014). In a new approach by Burrows et al. (2014) the parameterization/ OM prediction for marine aerosol particles is based on important compound classes of OM, e.g. lipids, carbohydrates, proteins, humic-like compounds, instead of chl-$a$ concentrations in seawater. To apply and further develop such OM parameterizations/ predictions, distinct measurements of these specific organic compound groups on molecular level in different oceanic regions are urgently needed. To this end, concerted measurements such as those we have performed in this study are essential. With studies like ours, simultaneous measurements of e.g. lipid classes in both marine compartments (seawater and aerosol particles) are obtained, and such data can finally be used to improve organic matter transfer models.

In the revised manuscript, we summarized the current state of knowledge regarding chl-$a$ and OM parameterizations now more clearly (page 3, line 22-30): "Most parameterizations, the transfer of OM and the prediction of the OM content on marine aerosol particles, is based on chl-$a$ seawater concentrations that are used as a broad indicator of biological productivity (Gantt et al., 2011;Rinaldi et al., 2013). However, especially in oligotrophic regions additional parameters besides wind speed and chl-$a$ must be taken into consideration for accurately prediction of OM on marine aerosol particles (van Pinxteren et al., 2017). In a new approach by Burrows et al. (2014) the parameterization/ OM prediction for marine aerosol particles is based on important compound classes of OM, e.g. lipids, carbohydrates, proteins, with different physico-chemical properties instead of chl-a concentrations of the seawater. This new approach requires information of distinct OM groups in both marine matrices (seawater and aerosol particles) such as the lipid concentrations performed in the present study."

R#2-2) In general, it is crucial to discuss location of sampling and the temporal succession of sample events since enrichment factors for aerosols are calculated and, more importantly, conclusions on processes leading to the observed enrichment are drawn. Marine sources of the aerosols sampled at CVAO most likely do not match the location where they have sampled the SML and ULW, nor did sampling of aerosols (over the course of 24 hours) matches the specific time slots of seawater sampling.

The reviewer addressed a very important point: the comparability of aerosol particle measurements and seawater measurements. In our approach of concerted measurements, we combined PM$_1$ aerosol particle samples (sampled for 24 h) with spot samples taken in the ocean (ULW, SML) during the aerosol sampling period. To allow a comparison of these two matrices, we strongly considered several additional parameters, such as backward trajectories, the concentrations of inorganic ions and mineral dust tracers on the aerosol particles measured during the campaign. These parameters were discussed

in detail in the overview paper of the campaign and in a paper regarding amino acid measurements within this campaign and show that aerosol particles were predominantly of marine origin with low to medium dust influences (Triesch et al., 2020;van Pinxteren et al., 2020). In this context a possible transport of the aerosol particles is also discussed.

We have addressed this briefly in a new subchapter '3.2.1 The comparability of the different marine matrices (seawater and aerosol particles)' of section '3.2 Transfer of lipids from the Oceans'. It reads now as follows (page 12, line 6-11): "The concerted measurements performed here included spot samplings in the ocean (ULW, SML) during the sampling period of $PM_1$ aerosol particles at the CVAO (24h). The air masses arriving at the CVAO often followed the water current (Peña-Izquierdo et al., 2012;van Pinxteren et al., 2017) and suggest an enhanced link between the upper ocean and the aerosol particles, as mainly winds drive the ocean currents in the upper 100 m of the ocean. The backward trajectories as well as the concentrations of inorganic ions and mineral dust tracers on the aerosol particles measured during the campaign, suggested a predominantly marine origin with low to medium dust influences (Triesch et al., 2020;van Pinxteren et al., 2020)."

R#2-3) The authors further compared the enrichments of lipid classes in the SML and aerosols to a defined theoretical 'surface activity' characterized by certain criteria i.e. density, partitioning coefficient between octanol and water (Kow) and topological polar surface area (TPSA). It should be kept in mind, that the solute is water and thus surface enrichment of lipids may be rather dictated by amphiphilic behavior i.e. an increase in TPSA and lower Kow.

We thank the reviewer for his comment. The 'surface activity` is a parameter regarding the enrichment of lipid classes in the SML and on the aerosol particles (e.g. Burrows et al. (2014)). To estimate and describe this surface activity for lipids we used different physico-chemical parameters ($K_{OW}$, TPSA and density) as described and discussed in detail in the SI (page 26/27) in section "surfactant activity of investigated individual lipid classes".
We agree with the reviewer that the amphiphilic structure of lipids in water is not negligible. Due to hydrophilic and hydrophobic components within the lipid structures, it is likely that not only one distinct parameter, but e.g. a combination of physico-chemical parameters might explain the observed enrichment of lipid classes on the aerosol particles. As suggested by the reviewer, we therefore performed a multilinear regression of the parameters $EF_{aer}$, TPSA and $K_{OW}$ as shown in the Figure R1 below. The statistical parameters of this regression ($f(EF_{aer})=k1+k2*TPSA+k3*K_{OW}$) are the following: $R^2=0.45$ and n=11 but p=0.0875, which can therefore not be described as statistically significant, but only as a trend.

[Figure]

*Figure R1: Plot of KOW, TPSA and the $EF_{aer}$ of the individual lipid classes*

Therefore, this approach unfortunately does not contribute any clear added value to the explanation of $EF_{aer}$ by these two physico-chemical parameters and we think it is not meaningful to include it in the manuscript.

R#2-4) Also, it should be discussed how the two models of 'surface activity' compare (the authors also calculate adsorption coefficients based on concentrations and saturation vapor pressure as proxy for enrichment in the bubble-water-interface). I am not yet completely convinced that these approaches actually help to understand surface dynamics of enriched substances in the marine realm.

The idea of the adsorption coefficient is based on the study of Kelly et al. (2004). In this study, they regarded the atmospheric distribution and enrichment of oxidized organic compounds in the aerosol particle from the gas phase to the aqueous phase. Assuming equilibrium state, we made the assumption that such a distribution and enrichment is also possible from the aqueous (seawater) phase to the gas phase. By considering these two distribution possibilities between gas and aqueous phase, we want to discuss the distribution of analytes in the seawater with respect to air bubbles in connection with the formation of primary aerosol particles by the bubble bursting process.
This is a new theoretical approach, where the observed differences in selective transfer of lipids (different $EF_{aer}$ for the different lipid classes) are described by the adsorption coefficients $K_a$ and $K_{aq}$, based on the individual physical parameters of the lipid classes (Henry's law constant (H) and saturation vapor pressure (p)). These constants are used to describe the physical-dynamic effects of the analytes in the liquid medium. By calculating the adsorption coefficient in air ($K_a$), for example, it is possible to determine from which analyte concentration the partial pressure of the analyte is exceeded. If this happens the analyte then condenses on existing surfaces. The adsorption coefficient in water ($K_{aq}$) expresses the maximum amount of the analyte that can be dissolved. If this value is exceeded ($K_a > K_{aq}$), enrichment takes place in this medium. As we have explained in the main text, with this theoretical approach the distribution of the analyte can be estimated:
"When $K_{aq} \gg K_a$ (Fig. S17a), the analyte should be preferred distributed (from water) to air (inside the bubble). When $K_a \gg K_{aq}$ (Fig. S17b) in turn, the analyte should be preferably distributed (from air) into water while the analyte should be preferred distributed within the bubble interface when $K_{aq} \sim K_a$ (Fig. S17c)." (Manuscript: page 17, line 14-15)
We agree that this theoretical approach needs to be further investigated in future studies (e.g. tank/laboratory studies), because of the low number of available data points. Still, we could show in this study that this theoretical approach has the potential be used to explain the variance of the $EF_{aer}$ for lipid classes.

In order to explain the adsorption coefficients and their significance more clearly, we have added the following statements to section 'Adsorption of the individual lipid classes at bubble air-water interface' on page 31 in the SI. It now reads as follows: "$K_a$ expresses the maximum gas-phase concentration of the analyte before condensation on surfaces occurs." (SI, page 31, line 16/17) and "$K_{aq}$ expresses the maximum amount of analyte that can be dissolved. If this value is exceeded ($K_a > K_{aq}$), enrichment takes place in this medium." (SI, page 31, line 19/20).

**2-5) In the end, I advise that the authors should focus on the biological context since all lipid classes seem to relate to the marine realm, degradation indices are derived, pigment analysis and basic abundances of microorganisms were measured and INP analysis ultimately shows that a strong biological component controls activity and aim to better link their findings.**

We agree that the biological link is very important in this study. We have emphasized this at several parts throughout the manuscript.

Starting with the introduction, we have mentioned that the information of lipid classes "can be used as specific markers for the identification of OM sources and biogeochemical cycles in the marine environment (Parrish et al., 1988;Frka et al., 2011)." (page 2, line 26/27)

Furthermore, we discuss which lipid classes have been reported in context to (micro)organisms (page 2, line 27- page 3, line 3). We have phrased one of the main objectives of this study as follows: „The present work aimed at investigating lipids at the Cape Verde Atmospheric Observatory (CVAO) as species representative for different lipid classes in the marine environment of the tropical Atlantic Ocean, to study their abundance, (biogenic) sources and selective transfer into the marine atmosphere. The lipid data set obtained for samples from different marine compartments at the CVAO is discussed with regard to its biological origin and its ice nucleation potential." (page 4, line 10-14)

In section 3.1.4 (manuscript page 11) not only the general biological results are introduced and discussed, but also associated in detail with individual lipid classes through statistical correlation analyses. Moreover, we concluded that "it is most likely that bacteria have influenced the lipid pool which is consistent with the results obtained from the lipid composition" (page 12, line 1-2).

The relationship between chemical, physical and biological measurement data was also considered and discussed as follows in section 3.3 (page 19, line 17-20): "The relationships presented here between the lipids in general and in particular the lipid classes with assigned biological context (PE, FFA) and INP activity at higher temperatures (-10 °C, -15 °C) in the ambient SML indicating that lipids in the tropical North Atlantic Ocean have the potential to contribute to (biogenic) INP activity when transferred to the atmosphere."

Finally, in the conclusion (page 19/20) we summarized our findings with a strong focus on biological lipid connections.

To this end, we believe that the connection between lipids and biological sources from the introduction through the results section to the conclusion is outlined through the manuscript, thus illustrating the connection between chemical and biological processes regarding lipids in the marine environment.

**Specific comments**

**2-6) Page 2 Line 24-26 Clarify: Marine dissolved lipids are produced either by dissolution from the particulate fraction, or 'by' primary production: : : living cells are also part of the particulate pool. Maybe better distinguish between abiotic and biotic processes and include the microbial loop?**

We agree with the reviewer's comment and clarified this sentence. Now it reads as follows (page 2, line 23/24): "Marine lipids can be produced by abiotic and biotic processes and play an important role as energy sources in the aquatic ecosystem (Parrish, 2013)."

**2-7) Page3 Line 3 Consider quoting Becker et al. 2018 on TG's as storage compounds in phytoplankton.**

We thank the reviewer for this comment and quoted the interesting paper you mentioned. This now reads as follows in the manuscript (page 2, line 32 - page 3, line 1): "Triacylglycerols (TG) indicate metabolic reserves (Frka et al., 2011) and are reported as storage compounds in phytoplankton (Becker et al., 2018)."

**2-8) Line 8-9 'However, Chl-a (concentration?) is also found to be a poor descriptor of autotrophs (biomass, cell abundance?), especially in oligotrophic regions (Quinn et al., 2014).' The authors should clarify this, since Quinn et al. concluded that Chl-a concentration is only a poor proxy for organic matter enrichment in aerosols.**

We agree with the reviewer's comment and removed this sentence. For a more detailed discussion and description of chl-*a* in seawater as a proxy for the prediction of OM on aerosol particles we would like to refer to the review comment R#2-1b).

**2-9) Line 27-30 ': : :TG lipid class serves as an indication that the aerosol particles consist to a certain extent of freshly emitted sea spray: : :' Additional literature or an explanation would be very helpful, since Schiffer et al. 2018 concluded that on SSA surfaces the 'reduction in activity could essentially reduce the processing (by BC Lipase) of triacylglycerols into fatty aicds' i.e. if TG is present in SSA, it is not necessarily an indicator of freshly emittance. Also relevant on page 13, line 8**

We thank the reviewer for this comment and agree that the transfer of TG lipase from the sea to the atmosphere mentioned in Schiffer et al. (2018) and their statement that lipases have the potential to change the composition of SSA (described in the study of Schiffer et al. (2018) as "Triacylglycerol lipases have recently been shown to be transferred from the ocean to the atmosphere in atmospheric sea spray aerosol (SSA). Lipases have the potential to alter the composition of SSA…") are not sufficient to make a statement about the freshness of the aerosol particles. Only under the assumption that the triacylglyercol lipase enzyme a) has been transferred to the aerosols and b) is actively present there, it is possible to say that TG can be degraded to FFA. However, these enzymatic investigations were not performed in this study. Therefore, we have removed these statements from the revised manuscript.

**2-10) Line 29 'In laboratory studies by the authors Schiffer et al. (2018), lipase enzymes have shown to be transferred from the ocean into the atmosphere: : :' Again, additional literature and explanations are needed. Schiffer et al. 2018 conducted a laboratory experiment on surface behavior of lipase and lipids in a Langmuir trough and conducted molecular dynamics simulations to judge on the activity of enzymes on SSA.**

We would like to refer to the previous reviewer comment R#2-9.

**2-11) Page 4 Line 19 A map illustrating seawater sampling stations and CVAO location including distances and height of tower (!) would be helpful. Also, wind directions over the sampling period seem crucial to your study.**

We agree with the reviewer's comment. We added a map illustrating the seawater sampling station and the CVAO including the distance between both stations and the height of the tower at the CVAO. Moreover, we added the prevailing wind direction during the sampling period in this map. The map is listed as Figure S1 in the Supporting Information (page 2).

**2-12) Page 5 Line 25-28 The authors should mention that the analysis was conducted by Flow Cytometry. It is not completely clear, if the analysis of eukaryotic (based on autofluorescence?) and prokaryotic cells (based on staining with SYBR green?) was conducted simultaneously and if autotrophic prokaryotes were excluded from prokaryotic cell numbers?**

According to the reviewer's suggestion, we provide more details about microbial cell counting via flow cytometry in the revised manuscript (page 6, line 7-14), which reads now as follows: "Microbial cell numbers were counted via flow cytometry after seawater samples were fixed, flash-frozen in liquid-nitrogen, and stored at -20 °C. For prokaryotic cells counts, all samples were stained with SYBR Green solution. Counting was performed after addition of latex beads serving as an internal standard. Further

details can be found in Robinson et al. (2019). Small autotrophic cells were counted in a separate measurement after addition of red fluorescent latex beads (Polysciences, Eppelheim, Germany). Cells were detected by their signature in a plot of red (FL3) vs. orange (FL2) fluorescence, and red fluorescence vs. side scatter (SSC). This approach allows discrimination between different groups of prokaryotic and eukaryotic autotrophs (Marie et al., 2010), which in our case were size classes defined as *Synechococcus-like* cells and *Nanoeucaryotes*."

As assumed by the reviewer, prokaryotic and eukaryotic counts were performed in two separate measurements, based on SYBR-green-staining and autofluorescence signals, respectively. Prokaryotic cell numbers (i.e. TCN) include autotrophic prokaryotes. We decided not to correct TCN for autotrophic prokaryotes, as the latter were 2 orders of magnitude less in abundance, and thus, their numbers had only negligible impact on TCN numbers and dynamics. Moreover, the correction of TCN would have needed proper discrimination of *Prochlorococcus-like* cells as well, which we dich not perform in the present study. In order to avoid confusion, we have deleted the wording 'heterotroph' for TCN (page 11, line 21), and added the above mentioned fact to page 11, line 32/33: "Due to the low abundance of *Synechococcus-like* cells, we assume that most bacteria counted as TCN are heterotrophic and could have taken up the 'metabolites'."

**2-13) Page 6 Line 21 ': : : while lower LI values indicate that the appearing lipid classes are very fresh or resistant to degradation: : :' In my opinion, this is somehow critical and should be explained in more detail, since degradation products are themselves defined by their resistance to further degradation. This influences also concluding remarks later on, e.g. Page 10, line 2 ': : :suggesting that the dissolved lipid classes were quite resistant to degradation: : :' How can the authors decide whether lower LI's indicates fresh production or resistance to degradation as introduced in the experimental section?**

Goutx et al. (2003) proposed to use this lipolysis index (LI) as a new tool which characterizes the degradation stage of labile organic matter in natural seawater samples. A dominance of ALC, FFA, MG, DG counterparts, i.e. lipids present in the living plankton, over lipid degradation product indicate lipid freshness. Therefore, higher LI means the lipids are more degraded and vice versa. We rewrote the sentence on page 7, line 5-7 and it reads now as: "Higher LI values are characteristic for enhanced OM degradation and metabolite release, while lower LI values indicate that the appearing lipid classes are more fresh or resistant to degradation."
Further, we would like to underline that lipid degradation indices (ALC, FFA, MG, DG), which can be detected by thin layer chromatography, are only first step in lipid degradation. They are subject to further degradation, whether to smaller molecules that are not any more soluble in organic solvents or converted to $CO_2$.

Moreover, we rewrote also the sentence on page 10, line 20 - page 11, line 2, which reads now as follow: "The LI of DL (Table S5) varied between 0.13-0.53 in the ULW and between 0.20-0.48 in the SML samples, suggesting that the dissolved lipid classes were somewhat more resistant to degradation."

Overall, we may assume that lipids that are found in the dissolved fraction were composed of more saturated compounds. It is known that saturated compounds are generally less reactive than unsaturated (e.g. Sun and Wakeham (1994).

**2-14) Line 13 'However, these differences between bacterial and phytoplankton sources are not reflected in the total observed (particulate) lipid pool, because degradation products like FFA also contribute strongly.' Since FFA are present in the particulate fraction they apparently had to be enclosed within intact cells or other larger particles (>0.7_m). To my understanding, FFA would be part of the dissolved fraction otherwise. Thus, I am**

not so sure if FFA can serve as an indicator of degradation when encountered within the particulate pool. Is the LI defined as a proxy for degradation in the dissolved and particulate phase likewise (Goutx et al. 2003)?

Lipids that are found in the dissolved fraction are part of non-living OM, and from free living bacteria. Those bacteria represent small part of all bacteria in the seawater. Therefore, we may assume that lipids from the dissolved fraction are of non-living origin. Moreover, FFA represent small fraction of total cell lipids, from low detection limit to 10% (Jónasdóttir, 2019). So, we take that it is reasonable to take FFA as and degradation product.
Concerning the LI: Besides Goutx et al. (2003), who suggested to use this lipolysis index (LI) as a new tool which characterizes the degradation stage of labile organic matter in natural sea water samples, the LI was also used by Parrish et al. (1995), who evaluated LI in the particulate fraction. Based on these previous studies, the LI can therefore be considered as a proxy for the degradation of OM in both the dissolved and the particulate fraction.

**2-15) Page 9 Line 15-16 The authors should quote, which lipid class they refer to when talking about 'chlorophyll degradation products'.**

In the manuscript we have now defined that 'chlorophyll degradation products' are the pigments (PIG) determined by the TLC-FID method.

This is now read as follows (page 10, line 15-17): "This coincides well with the low concentrations of chlorophyll degradation products (PIG), suggesting that only moderate grazing took place and the (pigment-containing) organisms were fresh and in healthy condition (van Pinxteren et al., 2020)."

**2-16) Line 18-19 Please clarify to what exactly you are referring to. Does 'This observation' relates to enhanced degradation in the SML or simply high LI values in the East Atlantic Ocean?**

We clarified this sentence and it now reads as follows (page 10, line 18-20): "A higher LI in the SML was also observed by Gašparović et al. (2014) in the East Atlantic Ocean and can be attributed to both bacterial and photochemical abiotic degradation (Christodoulou et al., 2009)."

**2-17) Page 10 Line 33 Since the authors judge on ': : :Chl-a as a proxy for bioproduction, may not sufficiently explain the variability of lipid classes: : :' They should introduce their results regarding Chl-a in greater detail instead of referring to a table in the supplementary material. Also, I do not recall an introduced scientific discussion concerning the reliability of Chl-a as a proxy for lipid classes.**

With regard to reviewer comment R#2-1b we want to emphasize again that chl-*a* was not used as a proxy for lipid classes. Therefore, we have removed this mentioned sentence from the manuscript. We have now included the results of the chl-*a* concentrations in the manuscript (with reference to the SI) in section 3.1.4.

This now reads as follows (page 11, line 9-11): "In addition, the chl-*a* concentration in seawater increased from 0.11 µg L$^{-1}$ to 0.60 µg L$^{-1}$ (Table S2) during the campaign, but was generally low compared to other subtropical/tropical regions or worldwide (Duhamel et al., 2019)."

**2-18) Page 11 Line 7-14 ': : : slightly higher enrichment of the particulate fraction: : :' I actually**

do not think, this is meaningful to discuss in relation to the presented results, since variance of the dissolved EF's range within the larger variance of particulate EF's and means only very slightly.

We agree with the reviewer's comment that this variance is not meaningful for such a discussion. Therefore, we have deleted the sentences dealing with the 'very slightly enrichment of the particulate fraction compared to the dissolved fraction of lipids' from the manuscript.

**2-19) Line 12-14 'Moreover, marine dissolved lipids can be produced by dissolution from the particulate fraction and through primary production and released during the life cycle and after cell death. This(!) might lead to a slightly higher SML enrichment of the particulate lipids.' Please elucidate, I cannot follow the conclusion made. Why does a dependence of the dissolved pool from the particulate pool indicate higher enrichments? Increased degradation and abiotic photochemical reaction within the SML could likewise produce higher enrichment of the dissolved fraction: : :**

With regard to the reviewer comment R#2-18, we have shortened the discussion on lipid enrichment in SML. In this context, we have removed these sentences from the manuscript.

**2-20) Page 15 32 Lead the reader towards your conclusion stating that 'a differentiation of the contribution' of the particulate versus the dissolved pool was not possible also when taking into account the size of the fractions. To my understanding it is more likely that the fraction of lipids smaller than 0.7_m (i.e. dissolved) contributed to submicron aerosols (PM1).**

Within the sample preparation procedure, the seawater was divided into dissolved fraction (<0.7 μm) and particulate fraction (0.7-200 μm). Aerosol particles (<1 μm) were collected on the $PM_1$ aerosol particles. It must be considered that in the bubble bursting process, the formed droplets are water drops, which gradually dries up and finally leads to the formation of the aerosol particles. Therefore, the size separation in seawater is not transferable 1:1 to the aerosol particles.
In our study, we calculated the $EF_{aer}$ based on the lipid concentrations of the dissolved fraction in seawater as well as of the particulate fraction. The enrichment factors were not very different in both cases and the conclusion are not affected. We discussed this in the SI (SI: page 28, line 10-13).

To underline this fact, we added in the revised manuscript (page 14, line 17-19): "The $EF_{aer}$ based on the particulate total lipids in SML was with an average of $2·10^5$ very similar to the $EF_{aer}$ of the dissolved total lipids ($3·10^5$) as discussed in the SI, Table S8."
As all the several lipid classes were present in the dissolved and particulate fraction, an attribution to the lipid classes on the aerosol particles to a dissolved or particulate seawater origin was not possible, as stated in the manuscript (page 17, line 2-3).

**2-21) Page 15 Line 14 I actually sense it is assumed that marine bacteria transmitted into the atmosphere behave similarly in terms of production and metabolism than within the hydrosphere i.e. their natural habitat. I think, this is a hypothesis which needs to be discusses more carefully. (Also Page 4, line 4)**

We apologize for this misunderstanding. We did not mean to imply that bacterial metabolism will be similar in both the 'original' aquatic habitat and in aerosols. This would be bay far too speculative and for sure we don't show any data to support this. We nevertheless think that microbial activity may

affect the composition of the OM pool of aerosol particles – either passively or actively – although the extent of either contribution is absolutely unclear.

We have rephrased this part to: "Here, besides passive contribution (i.e. providing lipids to aerosol particles upon cell disintegration), bacteria may also actively influence the OM composition of aerosols (i.e. lipid production or degradation). However, the extent of this passive and especially of potential active bacterial contribution to the lipid pool of aerosols warrants further studies." (page 16, line 19-21)

**2-22) Page 18 Line 19 ': : :samples are consistent with the results of Wilson et al. (2015) indicating that lipids in the tropical North Atlantic Ocean have: : :'. This could leave the reader under the impression that Wilson et al. 2015 have assessed lipids and concluded they contribute to the biogenic INP pool.**

We agree with the reviewer's comment and rephrased this sentence to avoid misunderstandings. It now reads as follows (page 19, line 17-20): "The relationships presented here between the lipids in general and in particular the lipid classes with assigned biological context (PE, FFA) and INP activity at higher temperatures (-10 °C, -15 °C) in the ambient SML indicating that lipids in the tropical North Atlantic Ocean have the potential to contribute to (biogenic) INP activity when transferred to the atmosphere."

**2-23) Line 34 'However, concentration of Chl-a, as often used proxy for biological production via phytoplankton, is not sufficient to describe lipid concentration.' Again, Chl-a is not described as a proxy to determine lipid classes in literature.**

We agree with the reviewer's comment and have deleted this ambiguously worded sentence. For a more detailed explanation of the relationship between chl-$a$ concentrations in seawater and the prediction of OM on the aerosol particles we would like to refer to the comment R#2-1b).

**2-24) Supplementary Material Page 28 Table S7 'XLogP3-AA' replaces Kow, which is found in the main text, yet for the method in use to calculate this value, no literature is provided.**

We have changed the designation in MS and SI and in the revised version accordingly and we continuously use (log) $K_{OW}$. The column in table S7 (page 26) has been renamed to log $K_{OW}$ and an explanation of the calculation has been added as a footnote, which reads as follows: "* The calculation of the octanol-water partition coefficient (KOW) is based on the XLOGP3-AA method, which predicts the log KOW as XLogP3-AA value of compound by using the known log KOW of a reference compound as a starting point (Cheng et al., 2007). For each compound we also used the PubChem database (https://pubchem.ncbi.nlm.nih.gov/), an open chemistry database at the National Institutes of Health (NIH), to extract chemical and physical properties."

**Technical comments**

**2-25) Page 1 Title: 'Concerted measurements of lipids in seawater and on submicron aerosol particles at the Cape Verde Islands: biogenic sources, selective transfer and high enrichments'. The authors should overthink the title, e.g. include instead of 'high enrichment', 'ice nucleating potential' to better describe the content of the article.**

We would like to point out that the correlations found between lipids and ice nucleation potential relate to seawater measurements. It is difficult to extrapolate a transfer into aerosol particles with respect to organic compounds and ice nucleation potential, since other possible aerosol particles sources must also be considered regarding aerosol particles with respect to INP activity. For example, dust can also play an important role for ice nucleating potential in the marine environment as discussed e.g. by Burrows et al. (2013). Because of these limitations, we consider the results from section 3.3 'Connection between lipids and INP activity in seawater' not strong enough to switch the focus of the paper and its title to the 'ice nucleating potential' of lipids.

We would therefore suggest to leave the title 'Concerted measurements of lipids in seawater and on submicron aerosol particles at the Cape Verde Islands: biogenic sources, selective transfer and high enrichments' (page 1) and also section 3.3 'Connection between lipids and INP activity in seawater' (page 18/19) as it is.

**2-26) Line 16-23 Exclude 'To this end'. The set of lipid classes analyzed includes : : : and rephrase the following sentence: Introduced lipid classes have been analyzed in the dissolved and particulate fraction of seawater, while differentiating between underlying water (ULW) and the sea surface microlayer (SML), and on submicron aerosol particles (PM1) collected from the ambient (air?) at the Cape Verde Atmospheric Observatory (CVAO). Or consider other fragmentation.**

We agree with the reviewer's comments and have reformulated the sentence as proposed and divided it into two sentences. It now reads as follows (page 1, line 16-24): "The set of lipid classes includes hydrocarbons (HC), fatty acid methyl esters (ME), free fatty acids (FFA), alcohols (ALC), 1,3-diacylglycerols (1,3 DG), 1,2-diacylglycerols (1,2 DG), monoacylglycerols (MG), wax esters (WE), triacylglycerols (TG), phospholipids (PP) including phosphatidylglycerols (PG), phosphatidylethanolamine (PE), phosphatidylcholines (PC), glycolipids (GL) including sulfoquinovosyldiacylglycerols (SQDG), monogalactosyl-diacylglycerols (MGDG), digalactosyldiacylglycerols (DGDG) and sterols (ST). Introduced lipid classes have been analyzed in the dissolved and particulate fraction of seawater, while differentiating between underlying water (ULW) and the sea surface microlayer (SML), and on ambient submicron aerosol particle samples (PM1) collected from the ambient at the Cape Verde Atmospheric Observatory (CVAO) applying concerted measurements."

**2-27) Line 24 Include ∑toalignstyletotherestofthetext**

We made sure the character ∑ was used uniformly throughout the manuscript according to the ACP Guidelines (Times New Roman in font size 10).

**2-28) Line 32-33: For aerosols, however, the high enrichment of lipids (as a sum) on aerosols corresponds well: : : Include 's' and exclude one of the redundant 'aerososls'.**

We included the 's' in aerosols and removed the redundant 'on aerosols' in this sentence. It now reads (page 1, line 32 - page 2, line 1) : "For aerosols, however, the high enrichment of lipids (as a sum) corresponds well with the consideration of their high surface activity, thus the $EF_{aer}$ (enrichment factor on submicron aerosol particles compared to SML) ranges between $9 \cdot 10^4$-$7 \cdot 10^5$."

**2-29 a)Line 32 Separate 'physico-chemical' to align style to the rest of the text.**

We used 'physico-chemical' in the whole text of the revised manuscript.

R#2-29b) Page 2 Keywords: consider to replace rather generic words such as 'seawater', 'concerted measurements', 'transfer' by e.g. 'sea surface microlayer', 'sea spray aerosols'to characterize the work.

Following the suggestion, we have revised the keyword list and it now reads as follows (page 2, line 13/14): "Lipids, organic matter, submicron marine aerosol particles (PM1), sea surface microlayer (SML), ice nucleating particles (INP), enrichment factor, concerted measurements, Cape Verde Atmospheric Observatory (CVAO)."

**2-30) Page 4 Line 6 Rephrase and clarify this sentence 'is discussed in terms of biological and physical (INP) parameters: : :' E.g. is discussed in the context of its biological origin and its ice nucleation potential.**

We rephrased and clarified this sentence accordingly. Now it reads (page 4, line 12/13): "The lipid data set obtained for samples from different marine compartments at the CVAO is discussed with regard to its biological origin and its ice nucleation potential."

**2-31) Page 5 Line 33 The authors should briefly explain the unit in use: Does the unit relates to the total filter area used for the extraction of lipids in aerosols (28.27cm2)?**

We briefly explained that the 28.27 $cm^2$ of the total filter area was used for the extraction of lipids in aerosol. It now reads as follows (page 6, line 20/21): "For the analysis of lipid classes, 28.27 $cm^2$ of the total $PM_1$ filter area were extracted and measured following the procedure for particulate lipids in seawater (see section 2.2.1)."

**2-32) Page 7 Line 14 Exclude, since this is a repetition of line 12: ': : :but considered a 'trend' to be valid: : :'.**

In section '2.2.5 Statistical analysis' we have discussed the statistical parameters that we used in seawater and the conditions for defining relationships statistically relevant or just as 'trends.' Since we have used both ULW and SML samples for the statistical analyses, but these have, for example, different sample numbers (n), which also affects the statistical parameters, we have defined both sample types individually. In order to make the selection criteria for a trend in ULW and SML (which are slightly different) easy for the reader to understand, we would therefore prefer to leave these explanations of the "trend in SML" in the manuscript.

**2-33) Page 8 Line 8 Maybe introduce the PE/PG ratio along with LI and EF's in the experimental section.**

We agree with the reviewer's comment and introduced the PE/PG ratio together with the LI in section 2.2.3 'Lipid ratios', which reads now as follows (page 7, line 9-12): "The PE/PG ratio can be used to determine the origin of the phospholipids that contribute to the OM pool in seawater (Goutx et al., 1993). Here, PG as the most important compound of the phospholipids of microalgae is an indicator for algae as potential sources, whereas PE is predominantly found in bacterial membranes and thus represents an indicator for bacterial sources (Goutx et al., 1993)."

**2-34) Line 10 Replace 'afterwards' with 'towards the end'.**

We took the suggestion and it now reads as follows (page 9, line 6-8): "The PE/PG ratio varied along the campaign with increasing values towards the middle of the campaign (maxima on 03/10/2017 and 04/10/2017) and decreasing values towards the end (Table S4), following the same trend as the total bacteria number (TCN, Table S3)."

**2-35) Line 11 Consider rephrasing or exclude '-': 'This indicates a change in the lipid dominant biological contributions, with bacterial sources dominating in the first part and especially in the middle of the campaign, whereas in the last part rather phytoplanktondominated contributions to the lipid pool.'**

We rephrased this sentence accordingly. It now reads as follows (page 9, line 8-11): "This indicates a change in the lipid dominant biological contributions, with bacterial sources dominating in the first part and especially in the middle of the campaign. However, in the last part of the campaign (afterwards on 05/10/2017) contributions to the lipid pool were rather dominated by phytoplankton."

**2-36) Page 9 Line 18 Include the articles ': : :release in the SML compared to the ULW: : :'.**

We inserted the articles 'the' and it now reads as follows (page 10, line 17/18): "However, on specific days, the LI$_{SML}$ of PL was ≥ 0.5 (Table S5), indicating increased OM/lipid degradation and metabolite release in the SML compared to the ULW."

**2-37) Page 10 Line 32 Exclude 'the': ': : :it is most likely that the bacteria have influenced: : :'**
Check for consequential mistakes.

We excluded 'the' and it now reads as follows (page 11, line 33 - page 12, line 2): "Although it remains unclear whether the bacteria have a passive (i.e. via membrane) or active (i.e. metabolism of the lipid 'metabolites') effect on the observed correlation between LI$_{PL}$ and TCN, it is most likely that bacteria have influenced the lipid pool which is consistent with the results obtained from the lipid composition."

**2-38) Page 11 Line 23 ': : :OM compound groups: : :' I think, this is redundant, use groups or compounds instead.**

We thank the reviewer for his comment and used 'OM compounds' in this sentence. Now it reads as (page 12, line 26-28): "A comparison of lipid enrichment with other OM compounds showed that SML enrichment of lipids seemed to be less pronounced in contrast to other organic species such as amino acids (Reinthaler et al., 2008; Triesch et al., 2020)."

**2-39) Line 28 Instead of 'regarded' use 'considered'.**

We agree and it now reads as follows (page 12, line 30-32): "It has to be considered that a SML described here represents a layer with a thickness of about 100 μm (van Pinxteren et al., 2017) and therefore gradients within this layer (e.g. an enhanced enrichment of surfactants only in the top layer of a few μm) cannot be considered here."

**2-40) Page 12 Line 6 ': : :different diversity and different taxa: : :' Redundant if diversity actually**

indicates the composition of species. However, it can be defined as functional etc.

We agree that this is redundant. We meant the overall community composition to differ between SML and ULW. This might imply differences in functional diversity as well and we corrected this sentence accordingly: "Besides increased abundance of microbial cells, this may also be due to a different microbial community composition between SML and ULW and thus, different functional diversity (Cunliffe et al., 2011)." (page 13, line 10-12)

**2-41) Line 8 ': : :in the particulate fraction. In the particulate fraction: : :' Try to rephrase due to repetition.**

We rewrote the second sentence to avoid repetition. It reads now as follows (page 13, line 12-14): "The metabolic reserves lipids, represented by TG, showed the highest variability of enrichment in the SML along the campaign in the particulate fraction. In this fraction, $EF_{SML(TG)}$ varied between 0.3 and 4.4, resulting in an averaged enrichment of 2.3."

**2-42) Line 11 Consider rephrasing: This indicates that the lipid reserves are stored in the particulate lipids and are dissolved producing dissolved TG. For example: This indicates that lipid reserves such as TG are stored within the particulate pool and upon dissolution become part of the dissolved pool.**

We rephrased this sentence as suggested. It now reads as follows (page 13, line 16/17): "This indicates that lipid reserves such as TG are stored within the particulate pool and upon dissolution become part of the dissolved pool."

**2-43) Line 13 Use 'physicochemical descriptors' instead of 'physical processes'.**

We agree and it now reads as follows (page 13, line 18/19): "Altogether, our results indicate that physicochemical descriptors alone, which are related to the surface activity of the lipids, are not sufficient to describe the SML enrichment of the lipids, at least not in the top 100 µm."

**2-44) Line 28 Caption of Fig. S11 states 'dissolved' lipids in aerosols particles, which is probably a mistake.**

We thank the reviewer for this comment. Regarding the reviewer's comment R#2-57, Figure S11 (percentage composition of lipids on aerosol particles) has been removed.

**2-45) Page 13 Fig. 3 Absolute concentration of lipids in aerosol particles do not fit percentage data in the supplement of Fig.S11. For example, on the 29/09/2017 PE are present in Fig.3 while being completely absent in Fig. S11, the color schemes might have been confused.**

We thank the reviewer for this comment. As correctly assumed, the color scheme of PG and PE was swapped in Figure S11 The uniform color scheme of the individual lipid classes in the Figures in the manuscript as well as in the Supporting Information was double-checked. PE now shows the defined red tone and PG the green tone, as is now the case throughout the paper (MS and SI). As mentioned above, regarding the reviewer's comment R#2-57, Figure S11 has been removed.

**2-46) Line 6 'bacteria, possibly transported from the ocean into the atmosphere, produce PE on aerosol particles': : : Better to replace 'produce' by 'contribute'.**

We agree with the reviewer's comment and replaced 'produce' by 'contribute' in this sentence. It now reads as (page 14, line 10-12): "Since heterotrophic bacteria are reported as a dominant source of PE (Michaud et al., 2018), this suggests that i) bacteria, possibly transported from the ocean into the atmosphere, contribute PE on the aerosol particles and/ or ii) PE is directly transferred from the ocean into the atmosphere, likely via bubble bursting."

**2-47) Line 10 Replace 'maritime samples' by 'of marine origin'.**

This sentence was removed during the revision of the manuscript.

**2-48) Page 15 Line 9 This is misleading, better state 'aerosol particles' instead of 'particle phase'.**

We agree with the reviewer's comment and used 'aerosol particles' instead of 'particle phase' to avoid any misunderstanding. It reads now as (page 16, line 10/11): "TG and ALC showed a high enrichment both in the SML and in the aerosol particles."

**2-49) Line 30 Rephrase: 'The finding here, that both DL and PL, contain similar classes of lipid, which are also found on the aerosol particles, suggest that both types of lipids in seawater are transferred to the aerosol particles via bubble bursting process.'**

We rephrased this sentence accordingly. Now it reads as follows (page 16, line 34 - page 17, line 2): "The finding that both classes of lipids (DL and PL) are found on the aerosol particles (Fig. 5, Fig. S12) indicates that both types of lipids can be transferred from seawater to the aerosol particles, e.g. via bursting the bubbles."

**2-50) Page 17 Figure 6 'interface' is hard to read. Improve color scheme. There is also a logical mistake, since the caption states 'Scheme of a bubble during the bubble bursting process'. During bubble bursting, the bubble actually has reached the air-water interface i.e. exhibits two surfaces oriented towards the air inside and the atmosphere outside. Otherwise, the caption should state 'during the process of a bubble rising through the water column': : :**

We improved the color scheme as suggested. Moreover, we rephrased the caption of this Figures. It reads now: "Figure 6: Scheme of a bubble during the process of a bubble rising through the water column, distinguished between 'air' (inside the bubble), 'water' (surrounding the bubble), the 'interface' (bubble surface) and the distribution of the lipid classes MGDG, TG and ALC related to their $K_a$ and $K_{aq}$ values". Figure 6 can be found on page 18 in the MS.

**2-51) Line 12 Rephrase 'contain: : : abilities'.**

We rephrased this sentence and it now reads as follows (page 18, line 12/13): "One main feature of biological components in general is their potential ability to contribute to ice nucleation and act as INP in the atmosphere (Šantl-Temkiv et al. (2019) and references therein)."

**2-52) Page 18 Line 11-12 Replace 'by: : :' with 'of sea spray aerosols'.**

We replaced 'by' with 'of'. Now it reads (page 19, line 11/12): "DeMott et al. (2018) reported that ice nucleation by particles containing long-chain fatty acids in a crystalline phase was relevant for freezing of sea spray aerosols."

**2-53) Line 25 Consider rephrasing: 'At the CVAO, concerted measurements of lipids as representatives of their respective classes were performed during the MarParCloud campaign to determine their concentrations in seawater and SML (as dissolved and particulate lipids) and on submicron aerosol particles.' For example: Concerted measurements of lipids were performed in proximity to the Cape Verde Islands to compare the concentration of specific lipid classes in submicron aerosol particles and in the dissolved and particulate phase of seawater (ULW and SML).**

We rephrased this sentence as suggested. Now it reads (page 19, line 25/26): "Concerted measurements of lipids were performed in proximity to the Cape Verde Islands to compare the concentration of specific lipid classes in submicron aerosol particles and in the dissolved and particulate fraction of seawater (ULW and SML)."

**2-54) Line 27 Consider rephrasing: E.g. The analysis of lipid classes in seawater showed that, although concentrations in the particulate and dissolved phase are generally very similar, the contribution of lipids within phases differed.**

We rephrased the sentence accordingly. It reads now as follows (page 19, line 27/28): "The analysis of lipid classes in seawater showed that, although concentrations in the particulate and dissolved fraction are generally very similar, the contribution of lipids within fractions differed."

**2-55) Page 23 Line 31-35 Check format, looks like a line spacing error.**

We corrected the line spacing error.

**2-56) Page 25 Line 1 Adjust the predicate 'Van' to the same format i.e. 'van'.**

We have now written 'van Wambeke' as well as 'van Pinxteren'.

**2-57) Supplementary Material I recommend to shorten the supplementary information provided, maybe consider excluding Fig. S7, 8, 11, S12, S13, S19.**

We have carefully reviewed the supplementary information and the proposed reductions and agree that in the submitted SI version Figure S7, S8 and S11 can be removed.
Regarding the other proposed figures in the SI, we would prefer not to exclude them as S12, S13 and S19 are important information for a better understanding of the context. Figure S12, 'Boxplot explanation related to Fig. 4', provided more information on the shown Boxplot (Fig. 4) in the MS. Figure S13, 'Correlation plot of the $EF_{aer}$ and the corresponding log KOW of the individual lipid classes: HC, TG, FFA, ALC, ST, 1,2DG, MGDG, DGDG, SQDG, PG, PE', showed the correlation and the $R^2$ of the relationship between the $EF_{aer}$ and the log KOW of individual lipid classes, which are discussed in the

MS. Figure S19, 'Overview of possible distributions of the analyte between interface, water and air: a) Kaq>>Ka, analyte is preferred distributed (from water) to air; b) Ka>>Kaq, analyte is preferred distributed (from air) into water; c) Kaq~ Ka, analyte is preferred distributed at the interface' illustrates the effects of the adsorption coefficients on the possible distribution of the analytes between interface, water and air and is in our opinion an important aid for the understanding of this adsorption coefficient approach.

By omitting the figures S7, S8 and S11 (so called in the submitted SI), the SI could be tightened and in addition there is a new numbering of the figures in the revised SI and therefore also of the references in the MS.

**2-58) Equalize color scheme and figures i.e. when drawing a regression line, use the same design and report same correlation values as in the main text e.g. R versus R2**

We unified the design of the regression lines accordingly. Figures S11 (SI, page 12) and S16 (SI, page 17) have been adapted to the design of the other regression lines. The regression lines are now shown in black and the corresponding $R^2$ is also noted in the graph. Figure S9b) and Figure S9c) (SI page 10) is still an exception. Since the regression lines for ULW and SML are shown in the same graph, the regression lines for ULW are shown in black and for SML in blue for a better visual differentiation.

**Additional changes performed by the authors**

The acknowledgement was also revised to thank the people from the OSCM. The added sentence is now as follows: "We further acknowledge the professional support provided by the Ocean Science Centre Mindelo (OSCM) and the Instituto do Mar (IMar)." (page 21, line 4-6)

The measured data were published on PANGAEA. The data availability statement was therefore updated and reads as follows: "Data availability. The data are available through the World Data Centre PANGAEA under the following link: https://doi.pangaea.de/10.1594/PANGAEA.921832." (page 20, line 27/28)

The previous citation of van Pinxteren et al. (2019) was updated to van Pinxteren et al. (2020) in the revised manuscript and supporting information

**References**

[revised manuscript text omitted]

---

## Author Response (AR2)

**Report#2**
Comments on the revised version of the work by Triesch et al., entitled "Concerted measurements of lipids in seawater and on submicron aerosol particles at the Cape Verde Islands: biogenic sources, selective transfer and high enrichments"
In the amended version of this work, I think that the Authors have done a good job to accomplish the comments/requests raised in the first round of review by two anonymous reviewers.
I have few additional comments that the Authors should consider to improve their work for publication.

We thank the reviewer for the careful examination of the manuscript and the supporting information. In the following, please find a point-by-point response to the questions and concerns. All references to the manuscript (e.g. page and line numbers) listed in our replies refer to the clean version of the revised manuscript (without track changes).

1. The text would benefit from a thorough revision by a native English speaker, to correct the syntax and phrasing of some sentences, make some of them more clear and sounding, check verb tenses (past or present? This should be more consistent throughout the text), correct some typos and avoid some improper/odd expressions.

Following this comment, the manuscript and supporting information has been carefully revised by a professional English-speaking person. The main focus was on the comprehensibility of the sentences/paragraphs.

2.a) Some doubts about statistics. I missed what was used as a test to assess significant differences in the comparisons among samples. This should be stated in the "2.2.5 Statistical analysis" section. Moreover, please avoid/correct too general expressions related to the results of the statistical tests performed.

We thank the reviewer for this comment and corrected the general expressions related to the results in the manuscript, please see also comment 2.b). Regarding the statistical analysis and their interpretations, we would like to refer to section 2.2.5. In this section, the terms 'statistical relevance' and 'trend' were defined. To validate the significance of the correlation, the correlation coefficient (R), the number of samples examined (n) and the p-value were used. In particular, the p-value as a test for statistical hypothesis in research areas must be considered when defining statistical relevance (Bhattacharya and Habtzghi, 2002;Perezgonzalez, 2015).

2.b) Here are only some examples from the text, but this aspect should be corrected throughout the text:

We agree with the reviewer and have specified the noted expressions and phrases below.

-"corresponds well", please be more specific
On page 2, line 1/2, it now read as: "On the aerosol particles, an EF$_{aer}$ (the enrichment factor on the submicron aerosol particles compared to the SML) between $9 \cdot 10^4$-$7 \cdot 10^5$ is observed."
On page 16, line 3-7 it read as follows: "In contrast to the SML enrichment, the higher enrichment of the lipids on the aerosol particles observed here corresponds to the high

surface activity of the lipids and the preferred adsorption to (bubble) surfaces resulting in a strong sea-to-air-transfer (Tervahattu et al., 2002;Facchini et al., 2008;Cochran et al., 2016a;Schmitt-Kopplin et al., 2012;Rastelli et al., 2017), and their possible association with other compounds promoting co-aerosolization processes (Quinn et al., 2015;Hoffman and Duce, 1976;Rastelli et al., 2017)."

-"becomes lower", quantitatively? Is it significant? (p value?)
-"increases", quantitatively? Is it significant ? (p value?)
This sentence has been carefully reworded and now reads as follows (page 8, line 22-26): "On sampling days when the PE concentration was high (e.g. 3/10/2017: 41.8 µg L$^{-1}$; 04/10/2017: 41.9 µg L$^{-1}$), the PG concentration, however, was lower (3/10/2017: 8.8 µg L$^{-1}$; 04/10/2017: 12.6 µg L$^{-1}$, see Fig. 1), whereas towards the end of the campaign, the concentration of PE decreased by a factor of 4-5 (e.g. 7/10/2017: 7.6 µg L$^{-1}$; 9/10/2017: 9.8 µg L$^{-1}$), while the concentration of PG (7/10/2017: 14.8 µg L$^{-1}$; 9/10/2017: 14.4 µg L$^{-1}$) increased by a factor up to 2."

-"showed not only a much lower concentration but also much less pronounced variance", please be more specific.. Is it significant? What do the Authors mean with "variance"? Did they perform a test based on the confidence intervals of different groups of samples?
The term 'variance' was intended to describe the smaller range of concentrations of TG (ULW: 0.9-5.8 µg L$^{-1}$; SML: 2.1-4.5 µg L$^{-1}$) and ST (ULW: 0.3-0.7µg L$^{-1}$; SML: 0.7-2.4 µg L$^{-1}$) compared to the higher concentrated FFA (ULW: 5.4-14.0 µg L$^{-1}$; SML: 16.1-36.5 µg L$^{-1}$) and PP (ULW: 15.2-54.9 µg L$^{-1}$; SML: 17.6-37.4 µg L$^{-1}$). This sentence has been carefully reworded and now reads as follows (page 8, line 19-21): "Within the PL, the lipid classes FFA (ULW: 5.4-14.0 µg L$^{-1}$; SML: 16.1-36.5 µg L$^{-1}$) and PP (ULW: 15.2-54.9 µg L$^{-1}$; SML: 17.6-37.4 µg L$^{-1}$) had high concentrations in seawater, while other lipid classes such as TG (< 5.8 µg L$^{-1}$) and ST (< 2.4 µg L$^{-1}$) had concentrations lowered by a factor of 4-23."

-"was always higher […], with one exception […]", this is incorrect phrasing
We agree with this comment and reworded the sentence, which reads now as follows (page 14, line 12/13): "However, although the atmospheric concentration of phospholipids was lower, PE was found to be more concentrated than PG, with only one exception on 27/09/2017 (Fig. 3)."

-"strong" or "strongly" or similar, please check all the times this word is (mis)used, and use a more sounding statistical/scientific wording
We checked the use of "strong" and "strongly" carefully and have changed the respective use as follows:
On page 4, line 1-3, it reads now as: "Cochran et al. (2016b) investigated the fatty acid composition on sub- and supermicron sea spray aerosol particles and reported that about 75 % of the submicron aerosol particles showed clear signals for the presence of long-chain fatty acids."
On page 8, line 31/32, it reads now as: "These differences between bacterial and phytoplankton sources are not reflected in the total lipid concentrations, because degradation products such as FFA also contribute to total lipids with a high proportion (Fig. 1)."
On page 12, line 34 – page 13, line 3, it reads now as: "A comparison of lipid enrichment with other OM compounds showed that the SML enrichment of lipids seemed to be less pronounced in contrast to other organic species such as amino acids (Reinthaler et al.,

2008;Triesch et al., 2021), despite the high surface activity of the lipids (Burrows et al. (2014), and references therein).”

On page 13, line 5-7, it now reads as: “The fact, that other (less surface active) compounds are more enriched in the SML (upper 100 µm) underlines the need to consider additional parameters to describe the SML enrichment of lipids in the ambient marine environment.”

On page 13, line 9-11, it now reads as: “Regarding the enrichment in the SML within the lipid classes or both fractions (the dissolved and particulate one), clearer differences were found when looking at the individual lipid classes.”

On page 14, line 7/8, it now reads as: “Compared to the seawater lipids, the atmospheric composition showed the same classes of lipids with an increased consistency of the DL composition (high contribution of HC and lower contributions of PP).”

On page 16, line 17-19, it now reads as: “Furthermore, a statistically relevant correlation ($R^2$=0.45, p=0.028) was found between the log $K_{OW}$ and the $EF_{aer}$ of the individual lipid classes (Fig. S11), indicating that the compounds with higher log $K_{OW}$ and thus a higher lipophilicity are preferably enriched on the aerosol particles.”

On page 20, line 11/12, it now reads as: “The fact that bacteria are clearly involved in lipid abundance underlines that models using chl-*a* are not enough to describe OM in general.”

-“weak relation”, is it significant or not? No sense to distinguish between stronger or weaker significance.

The correlation between log $K_{OW}$ and $EF_{aer}$ was found to be significant (R2=0.45, p=0.028). Therefore, the statement was specified in the abstract as follows (page 2, line 2-4): “Regarding the individual lipid groups on the aerosol particles, a statistically significant correlation ($R^2$= 0.45, p= 0.028) was found between $EF_{aer}$ and lipophilicity (expressed by the $K_{OW}$ value), which was not present for the SML.”

And in the conclusion (page 20, line 7-9): “In terms of the individual lipid groups on the aerosol particles, a statistically significant correlation ($R^2$=0.45, p=0.028) between $EF_{aer}$ and lipophilicity (expressed by the $K_{OW}$ value) was observed, which was not present for the SML.”

“somewhat more resistant to degradation” could the Authors be more “quantitative”?

Since the detailed investigation of the biotic and abiotic degradation of the individual lipid classes in both fractions was not part of this field study, only a comparison of the LI between the two lipid fractions is possible. In the future, for example, the degradation of the lipid classes should be investigated in more detail under controlled conditions in order to make a quantitative statement on this. We revised this sentence, which reads now as follows (page 11, line 9-11): ”The LI of DL (Table S5) varied between 0.13-0.53 in the ULW and between 0.20-0.48 in the SML samples, suggesting that the dissolved lipid classes were more resistant to degradation than the particulate lipids.”

“For aerosol, however, the high enrichment of total lipids corresponds well with the consideration of their high surface activity” this is an example of sentence, which is a bit odd and not clear in terms of statistical sense. Especially in the abstract and conclusions, the Authors should take particular care to use more sounding/scientific expressions.

We agree with the reviewer and rephrased this sentence. In the abstract it now reads as follows (page 2, line 1/2): “On the aerosol particles, an $EF_{aer}$ (the enrichment factor on the submicron aerosol particles compared to the SML) between $9 \cdot 10^4$-$7 \cdot 10^5$ is observed.”

**Additional corrections in the manuscript concerning too general expressions:**
The expressions "slightly" higher have been corrected and the sentences now reads as follows (page 9, line 6-9): "Compared to the particulate fraction, higher concentrations of total dissolved lipids were detected by a factor between 1.1 and 1.4 with $\sum$DL: 39.8-128.5 µg L$^{-1}$ in the ULW and with $\sum$DL: 55.7-121.5 µg L$^{-1}$ in the SML samples (Fig. 2). The maximum concentrations here were also a factor of 1.3-1.4 higher than the total dissolved lipid concentrations reported by Frka et al. (2011) in the Mediterranean semi-enclosed temperate Adriatic sea ($\sum$DL: 7.5-92.2 µg L$^{-1}$)."

The expression "significantly lower" has been corrected and the sentence now reads as follows (page 10, line 1/2): "Phospholipids, especially PE and PG, and FFA, which dominated the particulate lipids, showed lower concentrations by a factor of 1.1-2.1 within the total dissolved lipids."

The expression "slightly increased" has been corrected and the sentence now reads as follows (page 11, line 6/7): "However, on specific days, the LI$_{SML}$ of PL was $\geq$ 0.5 (Table S5), indicating an increased OM/lipid degradation and metabolite release in the SML compared to the ULW."

3. Regarding the comment by Rev#1: "R#1-8b) I don't know if it makes sense, but would it be possible to estimate an EFaer based on ULW properties instead of SML components?". The Authors' reply provides interesting information, but the Authors' conclusion is that this point raised by Rev#1 "does not provide any new insights […] and therefore we would prefer not to elaborate on this in the manuscript." Actually, I encourage the Authors to consider this point in the manuscript, as obtaining similar EFaer values if using ULW or SML values is not obvious and discussion about this could be deepened.

We thank the reviewer for his encouragement and have expanded the discussion regarding the EF$_{aer}$, to include the calculations with ULW as follows.
In section '2.2.4 Enrichment factors', the calculation of EF$_{aer}$ with respect to ULW has been extended as follows (page 7, line 27-33): "To calculate the enrichment factor of the different analytes on aerosol particles (EF$_{aer}$) relative to seawater (SW), here distinguished between SML and ULW, the atmospheric concentration of the analyte relative to the sodium concentration on the PM$_1$ sample was divided by the analyte concentration relative to the sodium concentration in the corresponding SW sample using equation (3):

$$EF_{aer} = \frac{c\,(analyte)_{PM_1}/c\,(Na^+)_{PM_1}}{c\,(analyte)_{SW}/c\,(Na^+)_{SW}} \qquad (3)"$$

In section "3.2.3 Transfer of lipid classes from the ocean to the aerosol particles", the discussion of EF$_{aer}$ was extended by comparing EF$_{aer}$ based on the ULW and based on the SML as follows (page 15, line 16-18): "The mean EF$_{aer(TL)}$ calculated based on the ULW concentration is $3.4 \cdot 10^5$, and thus very similar to the mean EF$_{aer(TL)}$ based on the SML concentrations ($2.6 \cdot 10^5$). This can be attributed to the fact that the lipid concentrations in the ULW and the SML were in the same concentration range, resulting in a comparatively low enrichment in the SML (EF$_{SML}$: 1.0-1.7, section 3.2.2)."

4. Figures. These could be improved in their graphical outputs (more consistent style - Avoid grey/unuseful lines - Use a heavier stroke to define bar/graph/axes contours – larger color legends – overall higher file resolution). Figure 6 might be a bit "poor" in information. This might be joined/enriched with one of the figures provided in the supplementary file (by transferring that in the main text). Or, just put Figure 6 in the supplementary file as well.

We thank the reviewer for his comment. We have improved the Figures as suggested, e.g. by adding larger color legends. Moreover, we removed the auxiliary line in Figure 4. In Figure 1-3 we would like to not completely omit the auxiliary lines to better understand the displayed concentrations when reading. In our experience, the graphics resolution of the Figures should not be a problem in the final upload of the figures. In addition, we have removed Figure 6 from the manuscript as suggested and now listed it as Figure S18 in the SI.

**Additional changes performed by the authors**

The previous citation of Triesch et al. (2020) was updated to Triesch et al. (2021) in the revised manuscript and supporting information.

**References**

Bhattacharya, B., and Habtzghi, D.: Median of the p Value under the Alternative Hypothesis, The American Statistician, 56, 202-206, 2002.

Perezgonzalez, J. D.: Fisher, Neyman-Pearson or NHST? A tutorial for teaching data testing Front Psychol, 6, 223-223, 10.3389/fpsyg.2015.00223, 2015.

Triesch, N., van Pinxteren, M., Engel, A., and Herrmann, H.: Concerted measurements of free amino acids at the Cape Verde Islands: High enrichments in submicron sea spray aerosol particles and cloud droplets, Atmos. Chem. Phys. Discuss., 2020, 1-24, 10.5194/acp-2019-976, 2020.

Triesch, N., van Pinxteren, M., Engel, A., and Herrmann, H.: Concerted measurements of free amino acids at the Cabo Verde islands: high enrichments in submicron sea spray aerosol particles and cloud droplets, Atmos. Chem. Phys., 21, 163-181, 10.5194/acp-21-163-2021, 2021.